# Distinct CCK-positive SFO neurons are involved in persistent or transient suppression of water intake

Takashi Matsuda [1,2], Takeshi Y. Hiyama[2], Kenta Kobayashi [3], Kazuto Kobayashi [4] &
Masaharu Noda [1,2✉]

The control of water-intake behavior is critical for life because an excessive water intake induces pathological conditions, such as hyponatremia or water intoxication. However, the brain mechanisms controlling water intake currently remain unclear. We previously reported that thirst-driving neurons (water neurons) in the subfornical organ (SFO) are cholecystokinin (CCK)-dependently suppressed by GABAergic interneurons under Na-depleted conditions. We herein show that CCK-producing excitatory neurons in the SFO stimulate the activity of GABAergic interneurons via CCK-B receptors. Fluorescence-microscopic $Ca^{2+}$ imaging demonstrates two distinct subpopulations in CCK-positive neurons in the SFO, which are persistently activated under hyponatremic conditions or transiently activated in response to water drinking, respectively. Optical and chemogenetic silencings of the respective types of CCK-positive neurons both significantly increase water intake under water-repleted conditions. The present study thus reveals CCK-mediated neural mechanisms in the central nervous system for the control of water-intake behaviors.

[1] Homeostatic Mechanism Research Unit, Institute of Innovative Research, Tokyo Institute of Technology, Yokohama, Kanagawa 226-8503, Japan. [2] Division of Molecular Neurobiology, National Institute for Basic Biology, Okazaki, Aichi 444-8787, Japan. [3] Section of Viral Vector Development, National Institute for Physiological Sciences, Okazaki, Aichi 444-8585, Japan. [4] Department of Molecular Genetics, Institute of Biomedical Sciences, Fukushima Medical University School of Medicine, Fukushima, Fukushima 960-1295, Japan. ✉email: noda.m.ae@m.titech.ac.jp

The physiological water balance in body fluids is maintained by the control of water intake and excretion[1]. Thirst has evolved for vertebrate terrestrial adaptation. Water-intake behavior in humans is influenced by several factors, including lifestyle, eating behavior, body temperature, and body-fluid conditions[2]. Water intake exceeding the capacity of water excretion causes hyponatremia (hypotonicity), while dehydration generates hypernatremia (hypertonicity) in body fluids[3,4]. Increased sodium levels ($[Na^+]$) in body fluids caused by the over-ingestion of salt also worsens the prognosis of cardiovascular diseases, such as hypertension[5] and cardiac failure[6,7]. These pathological conditions of body fluids lead to irreversible damage to organs, including the nervous system[8–10]. Thus, the neural control of water-intake behavior (thirst control) is crucial for maintaining life.

When animals lose systemic water, a sensation of thirst is aroused[2,11,12]. Water appetite causes animals to search for and drink water in order to restore the water-sodium balance to their physiological set points[12,13]. Previous studies demonstrated that thirst-driving systems are located in the forebrain sensory circumventricular organs (sCVOs)[2,11,13–16], such as the subfornical organ (SFO)[17] and organum vasculosum of the lamina terminalis (OVLT)[18], at which the blood–brain barrier is absent[14]: The SFO and OVLT are sensing sites for multiple thirst-driving signals, such as hypertonicity, sodium concentrations, and circulating angiotensin II (Ang II) in body fluids[19–21]. Local injections of Ang II into the SFO and OVLT, in which Ang II receptor 1a (AT1a) is positive, immediately elicited water drinking[21–24].

Regarding the SFO, we identified two neuronal subsets with AT1a that drive thirst (water neurons) and salt appetite (salt neurons), respectively[25]. Both neurons were positive for vesicular glutamate transporter 2 (Vglut2)[25] and neuronal nitric oxide synthase (nNOS)[25,26], indicating that they are excitatory neurons. We demonstrated that an increase in plasma Ang II levels caused by dehydration or salt depletion was sensed by AT1a in water neurons and salt neurons, respectively[25]. We also showed that the activity of water neurons in the SFO was suppressed by GABAergic neurons under Na-depleted conditions in a manner that was dependent on cholecystokinin (CCK)[25]; however, the CCK-mediated neural mechanisms underlying the inhibitory control of water intake have not yet been elucidated in detail.

CCK was originally identified as a gastrointestinal peptide hormone that controls food intake through the activation of visceral afferents[27]. CCK is one of the most abundant neuropeptides expressed in the brain, and is present at high levels in the hippocampus, amygdala, septum, olfactory tubercles, caudate nucleus, and hypothalamus[28]. CCK is intimately involved in a number of normal behaviors, such as learning, memory, feeding, nociception, and satiety[29].

In the present study, we identified distinct types of CCK-positive neurons in the SFO using CCK-ires-Cre (CCK-Cre) mice[30], and successfully characterized two types of CCK neurons using in vivo calcium imaging to monitor the dynamics of these neurons: one is persistently activated under hyponatremic conditions, while the other is transiently activated in response to water drinking. Electrophysiological investigations of these neurons showed that CCK neurons regulate the activities of GABAergic neurons in the SFO through CCK-B receptors. Our optogenetic and chemogenetic manipulations of these neurons revealed two distinct brain mechanisms for the persistent or transient suppression of water intake.

## Results

### CCK-positive neurons reside in the SFO
We previously demonstrated that CCK levels in the SFO markedly increased under Na-depleted conditions; however, the location of CCK-producing cells remained unclear[25]. To clarify whether CCK is derived from peripheral organs and delivered by the bloodstream, we initially measured CCK levels in plasma and the SFO. Blood and SFO tissues were collected from mice under the water- or Na-depleted condition by water deprivation or a furosemide treatment with a Na-deficient diet and water, respectively. Plasma CCK levels were slightly lower (0.7-fold) under the water-depleted condition that induced $[Na^+]$ elevations in body fluids than under the control condition (Na-repleted) (Fig. 1a, left). However, they remained unchanged under the Na-depleted condition in which $[Na^+]$ in body fluids was decreased (Fig. 1a, left). On the other hand, CCK levels in the SFO were markedly higher under the Na-depleted condition than under the control or water-depleted condition (9.0- and 9.4-fold, respectively; Fig. 1a, right), which is consistent with our previous findings[25]. These results suggested that increased CCK levels in the SFO under the Na-depleted condition were derived from the brain.

To clarify whether upregulated CCK is generated in the SFO itself, we stereotaxically injected an adeno-associated virus (AAV) carrying double-loxP-flanked inverted orientation (DIO)-mCherry (AAV-DIO-mCherry) into the SFO of CCK-Cre mice (Fig. 1b). A subset of cells expressing mCherry proteins in the SFO (namely CCK-positive cells) overlapped with those that were largely positive for nNOS ($89.4 \pm 2.88\%$ in CCK-positive cells, $n = 5$ sections) (Fig. 1c), but did not overlap with those that were positive for the glial fibrillary acidic protein (GFAP), a glial marker (Fig. 1d). Taken together with the previous finding that nNOS-positive cells in the SFO largely overlapped with the glutamatergic marker[26,31], this result indicates that these CCK-positive cells are excitatory neurons in the SFO; $21.9 \pm 1.86\%$ of nNOS-positive cells were positive for CCK. These mCherry-positive neurons did not express AT1a, an Ang II receptor, in the SFO of CCK-Cre; AT1a$^{lacZ/+}$ mice (Fig. 1b, e). Furthermore, CCK mRNA levels in the SFO were significantly higher under the Na-depleted condition than under the control condition (2.1-fold; Fig. 1f). Collectively, these results indicate that CCK upregulated in the SFO is produced by CCK-positive neurons within the SFO itself.

To exclude the possibility that CCK was also provided by neurons projecting to the SFO from other brain regions, we injected a highly efficient retrograde gene-transfer lentiviral vector (HiRet; ref. [32]) carrying DIO-mCherry into the SFO of CCK-Cre mice (Supplementary Fig. 1a). mCherry-labeled neuronal cell bodies were mainly present in the SFO, and slightly in the superior subnucleus of the lateral parabrachial nucleus (LPBN)[33], but were absent in the nucleus of the solitary tract (NTS)[34] (Supplementary Fig. 1b–d). However, CCK-positive LPBN neurons projecting to the SFO did not express Fos proteins under the Na-depleted condition (Supplementary Fig. 1e), suggesting that these CCK-positive neurons did not contribute to the increase in CCK levels observed in the SFO under the Na-depleted condition.

### GABAergic neurons are innervated by CCK-positive neurons in the SFO
We subsequently generated CCK-Cre; GAD67-GFP mice by crossing CCK-Cre mice with glutamic acid decarboxylase 67 (GAD67)-GFP mice to simultaneously visualize CCK-positive neurons and GABAergic neurons (Fig. 2a). When the SFO of CCK-Cre; GAD67-GFP mice was injected with AAV-DIO-mCherry, mCherry signals in CCK-positive neurons did not overlap with GFP signals in GABAergic neurons (Fig. 2b). However, mCherry-positive fibers and their varicosities of CCK-positive neurons were in close apposition to GABAergic neurons (Fig. 2b). These results are consistent with our previous findings

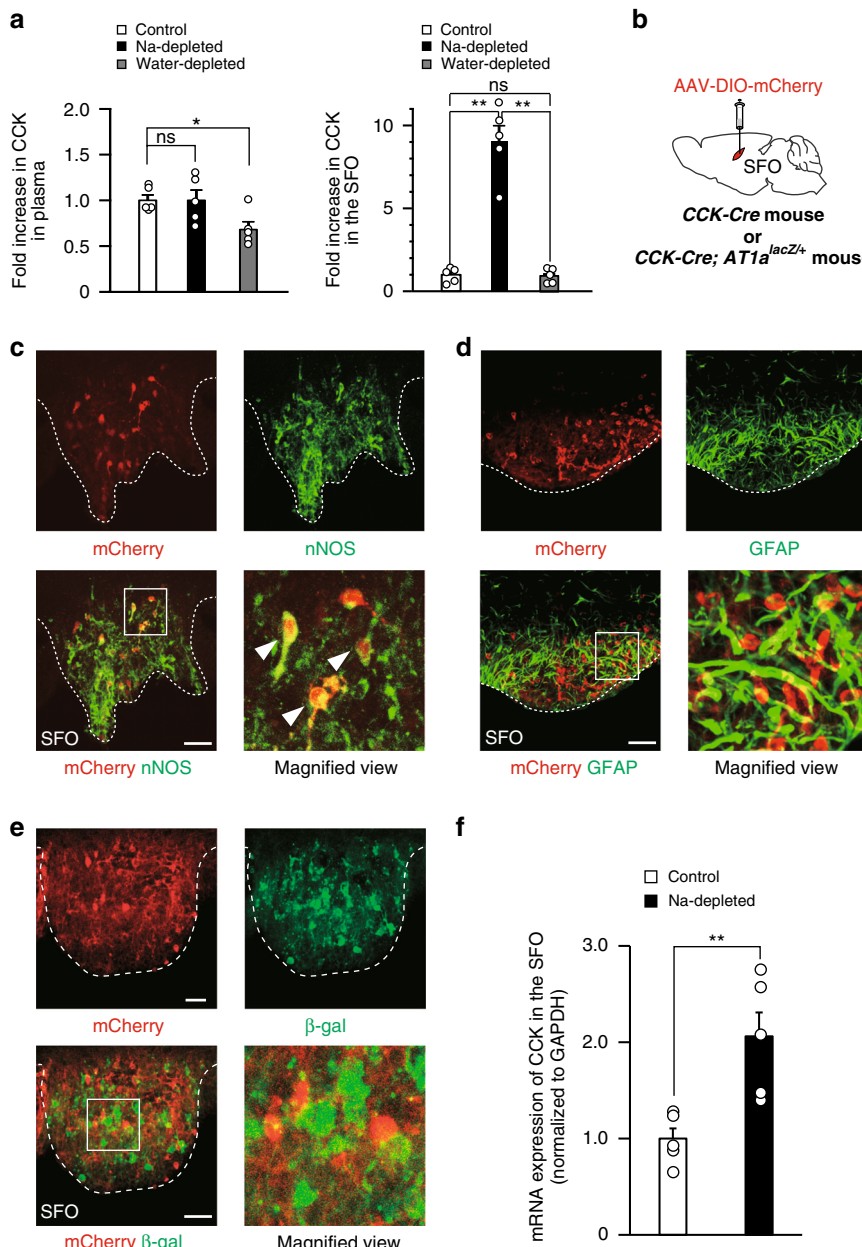

**Fig. 1 CCK-positive neurons in the SFO are AT1a-negative excitatory neurons. a** CCK levels in plasma (left) and the SFO (right) of mice under control, Na-depleted, and water-depleted conditions ($n = 5$ samples each; left graph, $U_{(Na)} = 12$, $P = 0.917$; $U_{(Water)} = 22$, $P = 0.047$; right graph, $U_{(Na)} = 25$, $P = 0.009$; $U_{(Water)} = 15$, $P = 0.676$; $U_{(Na\ vs.\ Water)} = 25$, $P = 0.012$). The right figure was reproduced from ref. [25]. **b** Injection of AAV-DIO-mCherry into the SFO of the *CCK-Cre* or *CCK-Cre; AT1a^{lacZ/+}* mouse. **c–e** Immunohistochemical detection of mCherry and nNOS (**c**), GFAP (**d**), and β-gal (**e**) in the SFO. The squared area in the merged panel was shown as the magnified view. Arrowheads indicate double-positive cells. These images were reproduced from more than three independent mice. **f** mRNA expression of CCK in the SFO under the control and Na-depleted conditions ($n = 5$ samples each; $U = 0$, $P = 0.008$). Scale bars, 50 μm. **$P < 0.01$; *$P < 0.05$; ns, not significant; two-sided Mann–Whitney $U$ tests in (**a**) and (**f**). Data represent the mean ± s.e.m.

showing that the activity of water neurons in the SFO was suppressed by GABAergic neurons in a CCK-dependent manner[25].

To confirm these results, we conducted electrophysiological experiments using acute brain slices containing the SFO: brain slices were prepared from *CCK-Cre; GAD67-GFP* mice in which the SFO had been injected with AAV-DIO-mCherry. In these slices, GABAergic and CCK-positive neurons were identified by green (GFP) and red (mCherry) fluorescence, respectively (Fig. 2c). In a subset of GABAergic neurons innervated by CCK-positive neurons (6/14 cells), we observed excitatory postsynaptic currents (EPSC), which were blocked by the AMPA glutamate receptor antagonist, CNQX (Fig. 2c). The firing

activities of these GABAergic neurons were stimulated by CCK, and suppressed by the CCK-B receptor antagonist, L-365,260 (Fig. 2d). Furthermore, $53.1 ± 6.52\%$ of GABAergic neurons in the SFO were positive for CCK-B receptors (Fig. 2e).

To verify that the activation of GABAergic neurons in the SFO suppressed water intake, we performed the optical activation of GABAergic neurons in the SFO using *vesicular GABA transporter (Vgat)-Cre* mice injected with AAV carrying DIO-channel rhodopsin 2 (ChR2)-EYFP into the SFO (Supplementary Fig. 2a). The optical excitation of GABAergic neurons in the SFO under the water-depleted condition, which strongly induced thirst, suppressed water intake significantly more than the no optical

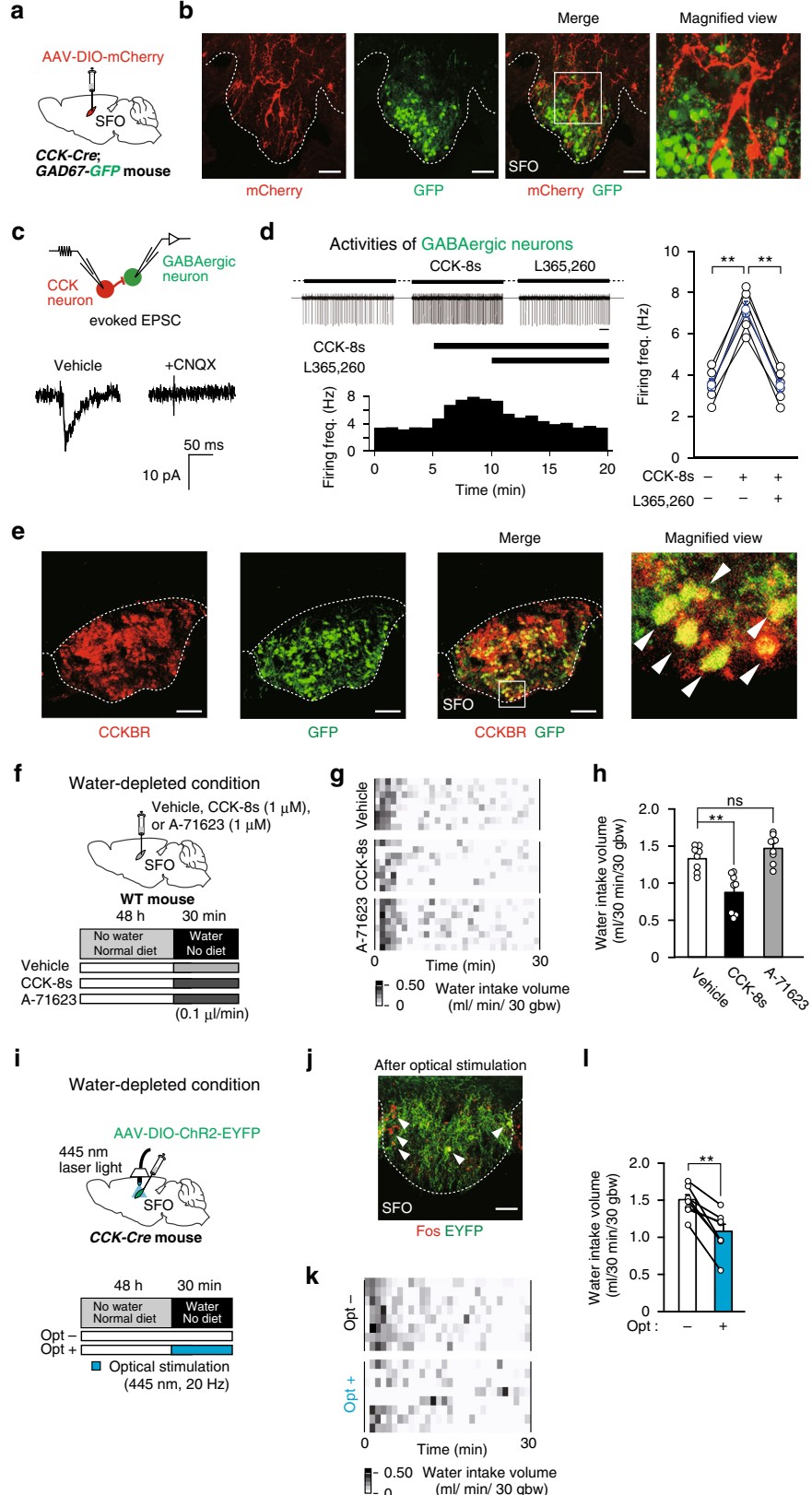

stimulation condition (1.40 ± 0.14 ml vs 0.84 ± 0.12 ml in 30 min; Supplementary Fig. 2b–d).

We then investigated the effects of CCK on water-intake behavior using the direct injection of the sulfated octapeptide of CCK (CCK-8s) into the SFO under the water-depleted condition (Fig. 2f and Supplementary Fig. 3a, b). Dehydrated mice with the

vehicle injection showed the normal enhancement of water intake (1.33 ± 0.06 ml in 30 min; Fig. 2g, h). As was expected, the CCK-8s injection into the SFO significantly reduced water intake to 0.87 ± 0.09 ml (Fig. 2g, h), which was equivalent to that in mice with the optical excitation of GABAergic neurons in the SFO (Supplementary Fig. 2d). Of note, the reducing effects on water

**Fig. 2 Activation of CCK-positive neurons in the SFO suppresses water intake through GABAergic interneurons expressing CCK-B receptors. a** Injection of AAV-DIO-mCherry into the SFO of the *CCK-Cre; GAD67-GFP* mouse. **b** Immunohistochemical detection of mCherry and GFP proteins in the SFO. The magnified view shows a representative image of mCherry-positive fibers and varicosities in close apposition to GABAergic neurons. Scale bar, 50 μm. **c** Upper: Schematic drawing of the electrical stimulation of a mCherry-labeled CCK-positive neuron (red) and whole-cell patch-clamp recording from a GFP-labeled GABAergic neuron (green). Lower: Representative excitatory postsynaptic currents (EPSC) in the GABAergic neuron evoked by the stimulation of the CCK-positive neuron. CNQX (10 μM) was added to the perfusate. The holding potential of the GABAergic neuron was −20 mV. **d** Left: Representative firing activities of GABAergic neurons recorded by the patch-clamp method in the cell-attached configuration. Effects of L-365,260, a CCKBR inhibitor, on the firing activity of GABAergic neurons in response to CCK-8s ($n = 6$ cells each; $W_{(CCK-8s)} = 0$, $P = 0.005$; $W_{(L365,260)} = 0$, $P = 0.005$). CCK-8s (0.1 μM) and L-365,260 (0.1 μM) were added to the perfusate during the time indicated with the horizontal thick line. Bar, 1 s. Right: Summary of the data obtained from 4 to 5 min, 9 to 10 min, and 14 to 15 min. **e** Immunohistochemical detection of CCKBR and GFP in the SFO of the *GAD67-GFP* mouse. Arrowheads indicate double-positive neurons in the magnified view. Scale bar, 50 μm. **f** Injection of CCK-8s (1.0 μM) and A-71623 (1.0 μM) into the SFO, and the experimental protocol under the water-depleted condition. **g** Grayscale heat maps of water intake by individual dehydrated mice with the vehicle, CCK-8s, or A-71623 injection. **h** Summary of data on water intake with the vehicle, CCK-8s, or A-71623 injection ($n = 8$ mice each). **i** Injection of AAV-DIO-ChR2-EYFP into the SFO of the *CCK-Cre* mouse, and experimental protocol for water intake with (opt+) or without (opt–) the optical stimulation (wavelength, 445 nm) under the water-depleted condition. **j** Immunohistochemical detection of Fos and EYFP in the SFO after the optical stimulation. Arrowheads indicate double-positive cells. Scale bar, 50 μm. **k** Grayscale heat maps of water intake by individual dehydrated mice with or without the optical stimulation. **l** Summary of water intake under the water-depleted condition with or without the optical stimulation ($n = 8$ mice each; $W = 36$, $P = 0.008$). Immunohistochemical images in (**b**), (**e**), and (**j**) were reproduced from more than three independent mice. bw, body weight; *$P < 0.05$; **$P < 0.01$; ns, not significant; two-sided Wilcoxon's signed-rank tests in (**d**) and (**l**); two-sided one-way ANOVA ($F(2,23) = 16.91$, $P = 4.20 \times 10^{-5}$) with post hoc two-sided paired Student's *t*-tests in (**h**) ($P_{(vehicle\ vs\ CCK-8s)} = 0.001$, $P_{(vehicle\ vs\ A-71623)} = 0.155$). Data represent the mean ± s.e.m.

intake by the CCK-8s injection were not detected under the normal condition because water intake in this condition was originally small over 30 min (Supplementary Fig. 3c–e). Next, we tested the effect of injection of a specific agonist for the CCK-A receptor (A-71623) into the SFO (Fig. 2f). The A-71623 injection under the water-depleted condition did not suppress water intake (Fig. 2g, h), supporting the view that CCK-B receptors, but not CCK-A receptors, are involved in the suppression of water-intake behavior.

We also examined the effects of the optical excitation of CCK-positive neurons in the SFO on water intake using *CCK-Cre* mice injected with AAV-DIO-ChR2-EYFP into the SFO (Fig. 2i). In comparison with the no optical stimulation condition, the optical stimulation of the SFO increased Fos expression levels in CCK-positive neurons (Fig. 2j) and significantly reduced water intake under the water-depleted condition (Fig. 2k, l). Collectively, these results indicate that a subpopulation of GABAergic neurons with CCK-B receptors relay inhibitory signals from CCK-positive neurons to water neurons in the SFO in order to suppress water intake.

**Two distinct subpopulations of CCK-positive neurons are present in the SFO**. To investigate the physiological activities of CCK-positive neurons in the SFO, we performed calcium imaging of these neurons at a single-cell resolution using miniature microscopy. The fluorescent calcium indicator GCaMP6f was conditionally expressed by the AAV-DIO-GCaMP6f injection into the SFO of *CCK-Cre* mice, and a gradient-index (GRIN) lens probe was inserted 200–300 μm above the SFO (Fig. 3a, b). We acquired Ca²⁺ fluorescence signals from individual CCK-positive neurons under the Na-depleted condition and hypertonic sodium-loading condition for a period of 30 and 60 s, respectively (Fig. 3c, d and Supplementary Fig. 4a, b). Consistent with the results shown in Fig. 1a, the majority of CCK-positive neurons were more strongly activated under the Na-depleted condition (Fig. 3e (i) and Supplementary Movie 1) than under the hypertonic sodium-loading condition (Fig. 3e (ii)). Of note, some CCK-positive neurons appeared to be activated at the onset of water intake induced by hypertonic sodium loading (Fig. 3e (iii) and Supplementary Movie 2).

A cluster analysis of the neural dynamics of individual neurons revealed that CCK-positive neurons may be classified into three distinct subsets (Fig. 3f). The largest population (group 1) showed stronger and more sustained activation under the Na-depleted condition than under the hypertonic sodium-loading condition, suggesting that the activity of this group reflects the body-fluid condition (Fig. 3g). In contrast, the second subpopulation (group 2) showed rapid and transient activation in response to water intake, and their activation stopped within ~20 s (Fig. 3f, g). More importantly, group 2 was not activated under the Na-depleted condition (Fig. 3g, h), suggesting that the activity of this population reflects water drinking because the body-fluid condition was not affected by the intake of a small amount of water in such a short period. The third subpopulation (group 3) did not show marked activation under the Na-depleted condition or the hypertonic sodium-loading condition (Fig. 3g, h). Thus, at least three distinct subpopulations of CCK-positive neurons may exist in the SFO, and the activities of two of these populations, group 1 and group 2, are controlled by different signals derived from [Na⁺] in body fluids and water ingestion, respectively.

**Activation of group 1 CCK-positive neurons suppresses water intake under the Na-depleted condition**. In order to confirm that the activity of group 1 CCK-positive neurons actually controls water-intake behavior, we performed the optical silencing of CCK-positive neurons in the SFO using a light-activated chloride pump (third-generation Natronomonas halorhodopsin; eNpHR3.0) under the Na-depleted condition, in which group 1 CCK-positive neurons are selectively activated (Figs. 3g, 4a, b). Na-depleted mice did not show additional water intake because of an imbalance between water and [Na⁺] in body fluids; however, the optical silencing of CCK-positive neurons significantly induced water intake under the Na-depleted condition (Fig. 4c). The amount of water intake induced by this treatment (0.31 ± 0.05 mL; Fig. 4c) was similar to the decrease observed in water intake with the optical activation of CCK-positive neurons under the water-depleted condition (0.43 ± 0.08 mL; Fig. 2l). Salt appetite, which is promoted under the Na-depleted condition, was not affected by the optical silencing of CCK-positive neurons (Supplementary Fig. 5).

We also examined the effects of the chemogenetic silencing of CCK-positive neurons on water intake under the Na-depleted condition. In these experiments, AAV-DIO-hM4D(Gi)-mCherry was injected into the SFO of *CCK-Cre* mice, and a subcutaneous injection of clozapine-n-oxide (CNO) was performed (Fig. 4d, e). The hM4D(Gi) is an engineered Gi protein-coupled receptor and

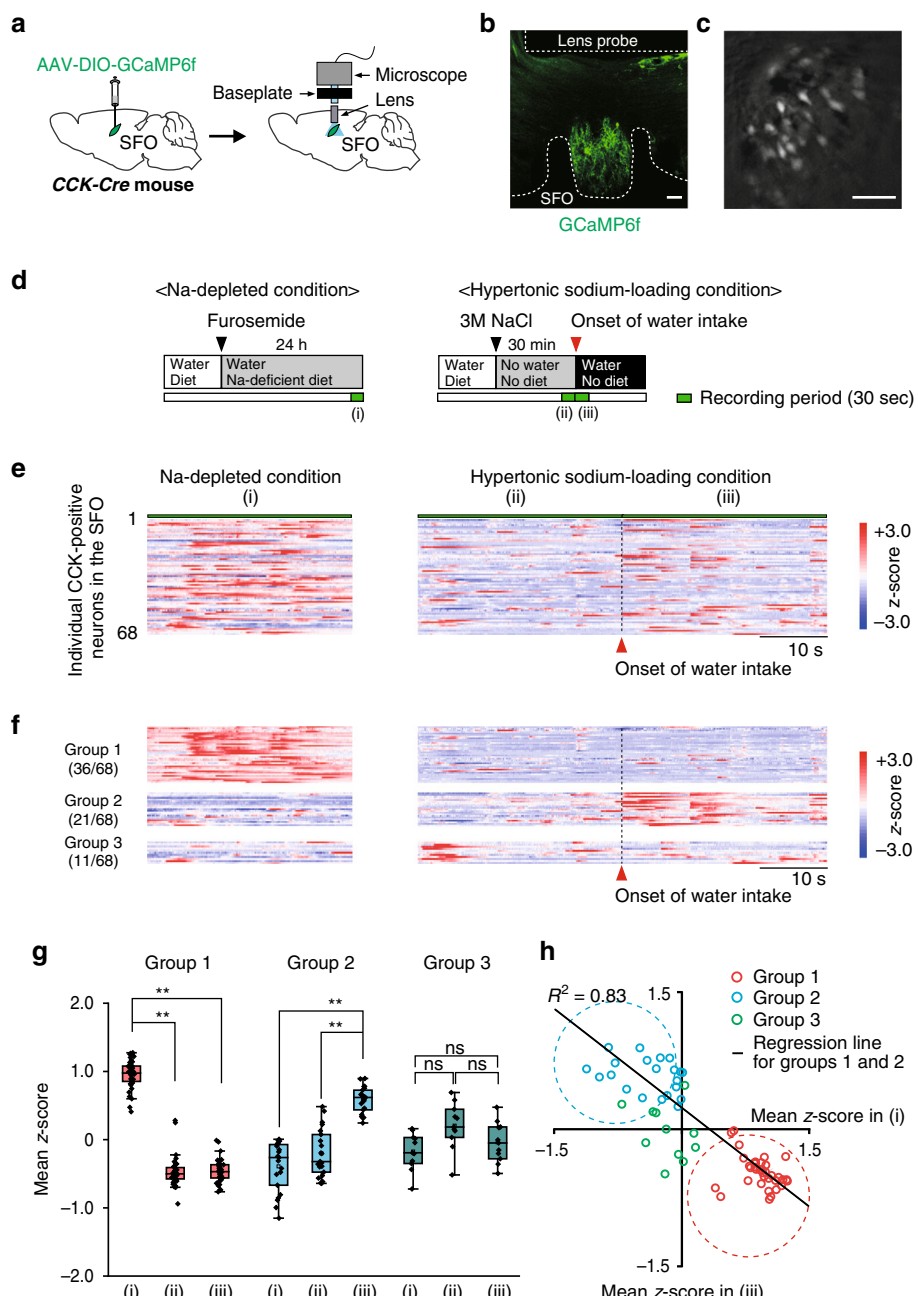

**Fig. 3 Distinct subpopulations of CCK-positive neurons are activated in the SFO under the Na-depleted condition or in response to water intake.**
**a** Injection of AAV-DIO-GCaMP6f into the SFO of the *CCK-Cre* mouse. GCaMP6f-expressing neurons were imaged through a GRIN lens targeting the SFO.
**b** Placement of the GRIN lens above the SFO and immunohistochemical staining of GCaMP6f in CCK-positive neurons in the SFO. Scale bar, 30 μm.
**c** Representative fluorescence-microscopic image of CCK-positive neurons. Scale bar, 30 μm. The images in (**b**) and (**c**) were reproduced from two independent mice. **d** Experimental protocols for $Ca^{2+}$ imaging of CCK-positive neurons (i) under the Na-depleted condition (left), (ii) under the hypertonic sodium-loading condition (right), and (iii) following the onset of water intake (right). **e** Raster plots of z-scored calcium responses in individual CCK-positive neurons during each condition ($n = 68$ neurons in two mice). **f** K-means clustering of individual CCK-positive neurons based on their activities under each condition. Three distinct subpopulations of CCK-positive neurons (group 1, group 2, and group 3) were identified in the SFO.
**g** Dot plot of the mean z-score of individual CCK-positive neurons in group 1 (left), group 2 (middle), and group 3 (right). $n = 68$ neurons in two mice. **$P <$ 0.01; ns, not significant; two-sided one-way RM ANOVA (group 1, $F_{(2,34)} = 380.9$, $P = 5.26 \times 10^{-10}$; group 2, $F_{(2,19)} = 97.17$, $P = 1.51 \times 10^{-10}$; group 3, $F_{(2,9)} = 2.524$, $P = 0.135$) with the post hoc two-sided paired Student's *t*-tests (group 1, $P_{(i\ vs\ ii)} = 1.15 \times 10^{-21}$, $P_{(i\ vs\ iii)} = 5.11 \times 10^{-25}$; group 2, $P_{(i\ vs\ iii)} = 6.68 \times 10^{-9}$, $P_{(ii\ vs\ iii)} = 6.58 \times 10^{-8}$). Box-plot elements (center line, upper and lower box limits, whiskers, and points) were defined as medians, quartiles, 1.5× interquartile ranges, and outliers. **h** Relationship between the mean z-scores of individual neurons under (i) the Na-depleted condition and (iii) the onset of water intake (linear regression between group 1 and group 2, $R^2 = 0.83$, $P = 6.77 \times 10^{-23}$). Red and blue dotted circles indicate the zones of group 1 and group 2, respectively.

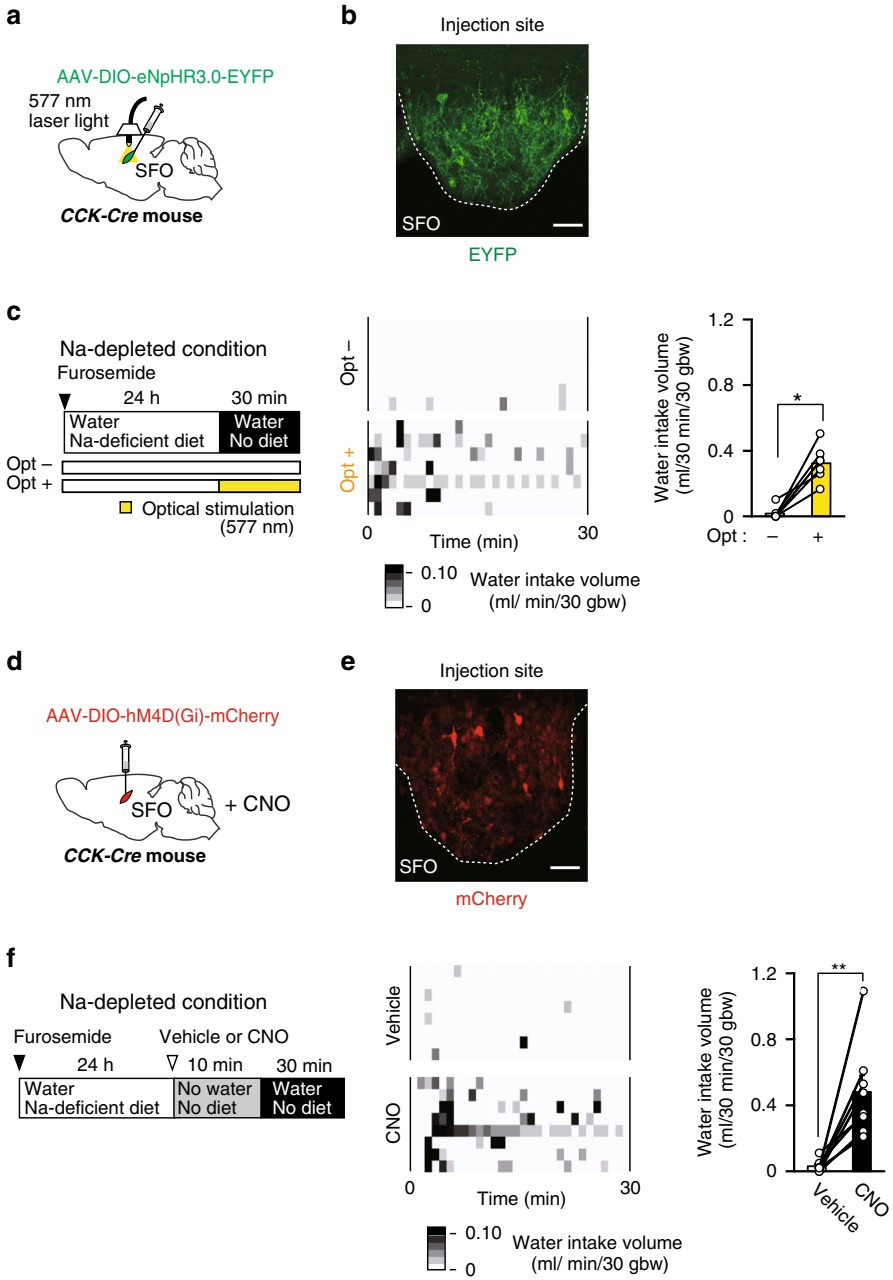

**Fig. 4 Optical or chemogenetic silencing of CCK-positive neurons in the SFO induces water intake under the Na-depleted condition. a** Injection of AAV-DIO-eNpHR3.0-EYFP into the SFO of the *CCK-Cre* mouse. **b** Immunohistochemical staining of EYFP in the SFO. Scale bar, 50 μm. **c** Left: Experimental protocol to stimulate water intake under the Na-depleted condition with (opt+) or without (opt−) optical inactivation (wavelength, 577 nm). Center: Grayscale heat maps of water intake by individual mice with or without the optical stimulation. Right: Summary of water intake under the Na-depleted condition with or without the optical silencing of CCK-positive neurons in the SFO ($n = 7$ mice each; $W = 0$, $P = 0.016$). **d** Injection of AAV-DIO-hM4D (Gi)-mCherry into the SFO of the *CCK-Cre* mouse. **e** Immunohistochemical staining of mCherry in the SFO. Scale bar, 50 μm. **f** Left: Experimental protocol to stimulate water intake under the Na-depleted condition by a subcutaneous injection of the vehicle or CNO (2 mg/kg). Center: Grayscale heat maps of water intake by individual mice treated with the vehicle or CNO. Right: Summary of water intakes with the vehicle or CNO injection under the Na-depleted condition ($n = 8$ mice each; $W = 0$, $P = 0.008$). Immunohistochemical images in (**b**) and (**e**) were reproduced from more than three independent mice. bw, body weight. **P < 0.01; *P < 0.05; two-sided Wilcoxon's signed-rank tests in (**c**) and (**f**). Data represent the mean ± s.e.m.

activated by designer drugs such as CNO. Activation of hM4D (Gi) results in neuronal inhibition for several hours. Similar to the optical silencing of CCK-positive neurons (Fig. 4c), chemogenetic silencing induced water intake under the Na-depleted condition (Fig. 4f): a previous study demonstrated that CNO exerted no effects on water intake by itself in WT mice[35]. Taken together, these results suggest that group 1 CCK-positive SFO neurons play an essential role in suppressing water intake under Na-depleted (or hydrated) conditions.

We focused on $Ca^{2+}$ fluorescence signals in group 1 CCK-positive neurons under the Na-depleted condition (Fig. 3g), and investigated whether the activities of these neurons were affected by the intake of salt solution (Fig. 5a). The strong activities of group 1 CCK-positive neurons under the Na-depleted condition

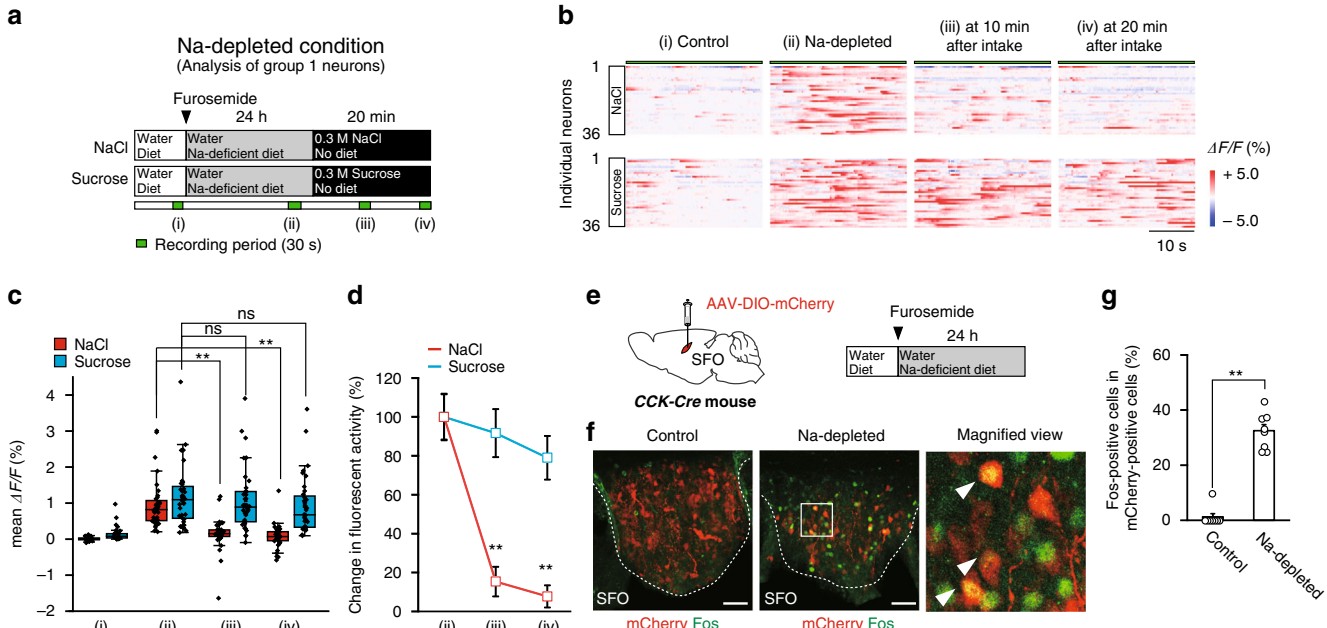

**Fig. 5 The activity of group 1 CCK-positive neurons in the SFO under the Na-depleted condition is attenuated by salt intake, but not fluid intake.**
**a** Experimental protocols for Ca$^{2+}$ imaging of group 1 CCK-positive neurons in the *CCK-Cre* mouse during 0.3 M NaCl or 0.3 M sucrose drinking under the Na-depleted condition. A large proportion of the intake of both fluids ceased within 20 min. **b** Raster plots of calcium responses in individual group 1 CCK-positive neurons under each sequential condition ($n = 36$ neurons in two mice). **c** Dot plot of calcium responses in individual CCK-positive neurons during 30 s at four temporal positions in (**b**). $n = 36$ neurons in two mice. ns, not significant; **$P < 0.01$; two-sided one-way RM ANOVA (NaCl, F(3,33) = 24.80, $P = 1.38 \times 10^{-8}$; sucrose F(3,33) = 33.87, $P = 3.47 \times 10^{-10}$), with post hoc two-sided paired Student's *t*-tests (NaCl, $P_{(ii\ vs\ iii)} = 9.84 \times 10^{-7}$, $P_{(ii\ vs\ iv)} = 9.21 \times 10^{-9}$; sucrose, $P_{(ii\ vs\ iii)} = 0.571$, $P_{(ii\ vs\ iv)} = 0.196$). Box-plot elements (center line, upper and lower box limits, whiskers, and points) were defined as medians, quartiles, 1.5× interquartile ranges, and outliers. **d** Changes in calcium responses from those under the Na-depleted condition. **$P < 0.01$; two-sided paired Student's *t*-tests ($P_{(iii)} = 3.47 \times 10^{-8}$; $P_{(iv)} = 3.24 \times 10^{-8}$). **e** Left: Injection of AAV-DIO-mCherry into the SFO of the *CCK-Cre* mouse. Right: Experimental protocol for the induction of the Na-depleted condition. **f** Immunohistochemical staining of mCherry and Fos in the SFO under the control and Na-depleted conditions. Arrowheads indicate double-positive cells under the Na-depleted condition. These images were reproduced from more than three independent mice. Scale bar, 50 μm. **g** Ratio of Fos-positive cells to mCherry-positive cells in the SFO ($n = 8$ sections each). **$P < 0.01$, two-sided Mann–Whitney *U* tests ($U = 0$, $P = 2.97 \times 10^{-4}$). Data represent the mean ± s.e.m. in (**d**) and (**g**).

markedly decreased after the start of salt ingestion (0.3 M NaCl) (Fig. 5b) and reached a plateau within 10 min (Fig. 5c and Supplementary Movie 1). In contrast, the ingestion of sucrose solution (0.3 M) did not significantly attenuate the activities of the CCK-positive neurons within 20 min, though the intake volumes of the sucrose solution were similar to those of the salt solution (Fig. 5b, c). The decreases observed in the activities of the CCK-positive neurons 20 min after salt ingestion were greater than those following sucrose ingestion (79.1 ± 11.2% vs 7.86 ± 5.71%; Fig. 5d). Decreases in the activities of these neurons following salt ingestion were not immediate or transient (no effects within 40 s) (Supplementary Fig. 6a–d). The number of Fos-positive CCK neurons markedly increased under the Na-depleted condition (Fig. 5e–g), indicating that the CCK-positive neurons are active when [Na$^+$] in body fluids is lower than the physiological set point.

**Group 2 CCK-positive neurons are involved in the tentative suppression of water drinking.** Recent studies showed that the activities of water neurons in the SFO of dehydrated mice were rapidly suppressed by the gastrointestinal stimuli of water ingestion[36–38]. Since group 2 CCK-positive SFO neurons are responsive to water intake, as described above, we investigated whether they are involved in this rapid suppression of water neurons. We performed the subcutaneous injection of a hypertonic NaCl solution (3.0 M NaCl) to induce water-intake behavior, and observed the activities of individual CCK-positive SFO

neurons during water drinking (Fig. 6a, b, Supplementary Fig. 7a, b, and Supplementary Movie 2). These neurons were not activated solely by licking the spout of an empty bottle; however, they were activated by the start of water drinking (Fig. 6b–d). The activities of group 2 CCK-positive neurons increased within 10 s, and then gradually decreased after 20 s (Fig. 6c, d). These CCK-positive neurons in the SFO were negative for the expression of Fos proteins under the water-depleted condition, but became positive after the resumption of water drinking (Fig. 6e–g), suggesting that the rapid suppression of water neurons is mediated by these CCK-positive neurons activated by the feedback signals of water intake.

We then confirmed that group 2 CCK-positive neurons that responded to water intake were also involved in the control of total water intake using optical and chemogenetic silencing. The optical silencing of these neurons was performed under the water-depleted condition, when group 1 CCK-positive neurons were not activated and group 2 CCK-positive neurons were activated in response to water drinking (Figs. 3g, 6c, 7a, b). Dehydrated mice showed markedly increased total water intake during the first 5 min after water was provided (Fig. 7c–e). The optical silencing of CCK-positive neurons significantly increased the volume of water intake during the first 5 min, but not in the subsequent 5–30 min (Fig. 7e, f), again indicating that group 2 CCK-positive neurons are relevant to the control of the amount of water intake.

We then performed the continuous inhibition of all CCK-positive neurons to examine whether the transient activation of

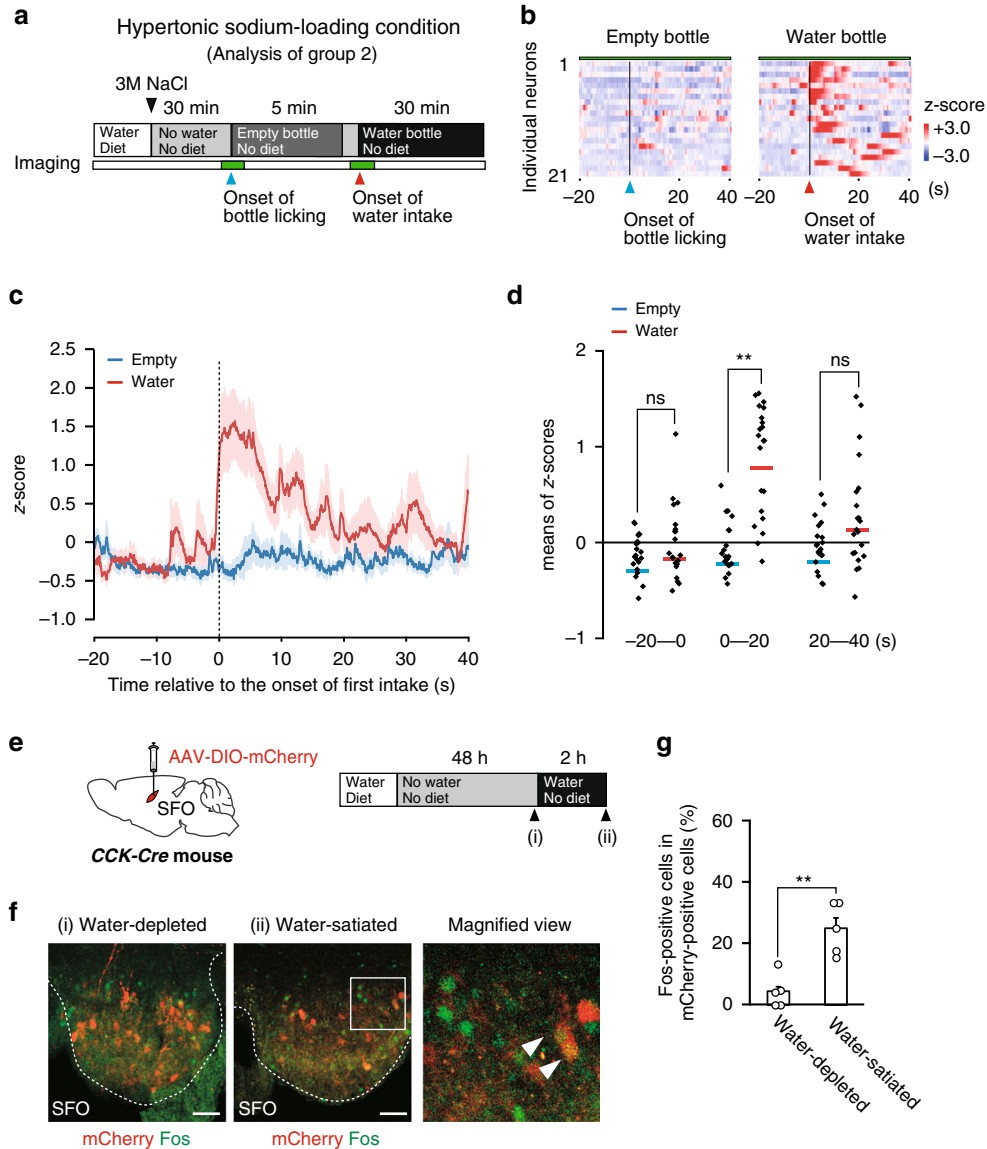

**Fig. 6 Group 2 CCK-positive neurons in the SFO are immediately activated in response to water intake. a** Experimental protocol for $Ca^{2+}$ imaging of group 2 CCK-positive neurons during drinking behaviors. **b** Raster plots of z-scored calcium responses by individual CCK-positive neurons during licking the empty bottle or drinking water ($n = 21$ neurons in two mice). **c** Averages of z-scored calcium responses during access to the water (or empty) bottle and drinking (licking). Shaded areas in red and blue indicate error bars. **d** Dot plot of mean z-scores of individual CCK-positive neurons during each 20-s interval at three temporal positions in (**c**). $n = 21$ neurons in two mice. ns, not significant. **$P < 0.01$; two-sided paired Student's t-tests ($P_{(-20)} = 0.333$; $P_{(0)} = 5.99 \times 10^{-6}$; $P_{(20)} = 0.053$). **e** Left: Injection of AAV-DIO-mCherry into the SFO of the *CCK-Cre* mouse. Right: Experimental protocol to confirm the activation of group 2 CCK-positive neurons after water intake under the water-depleted condition. **f** Immunohistochemical detection of mCherry and Fos in the SFO under the water-depleted and water-satiated conditions. Arrowheads in the magnified views indicated double-positive cells. These images were reproduced from more than three independent mice. Scale bar, 50 μm. **g** Ratio of Fos-positive cells in mCherry-positive cells in the SFO ($U = 0$, $P = 0.006$; $n = 5$ sections each). **$P < 0.01$; two-sided Mann–Whitney $U$ tests. Data represent the mean ± s.e.m.

group 2 neurons is involved in naturally occurring water intake under the normal (water- and Na-repleted) condition, in which group 1 neurons are not activated (Fig. 7g, h). During the experimental period (6 hrs from the start of the dark cycle), hourly water-intake volumes following the subcutaneous injection of CNO were significantly higher than those with the vehicle injection except for the first hour period (Fig. 7i–k): This is probably because it took a bit of time for the drug to show the effect. This result suggests that the activation of group 2 CCK-positive SFO neurons for the transient cessation of the thirst in response to respective water drinking is also involved in the control of total water intake under the normal condition.

## Discussion

We previously identified water neurons and salt neurons in the SFO that drive thirst and salt appetite, respectively[25]. Increases in $[Na^+]$ in body fluids under water-depleted conditions were sensed by $Na_x$ channels in the SFO, which consequently suppressed the activity of salt neurons through the activation of a subpopulation of GABAergic neurons[25,39,40]. On the other hand, water neurons were suppressed under Na-depleted conditions through the CCK-mediated activation of another subpopulation of GABAergic neurons[25]. However, the source of increased CCK in the SFO under Na-depleted conditions was unclear. In the present study, we demonstrated that CCK-positive excitatory

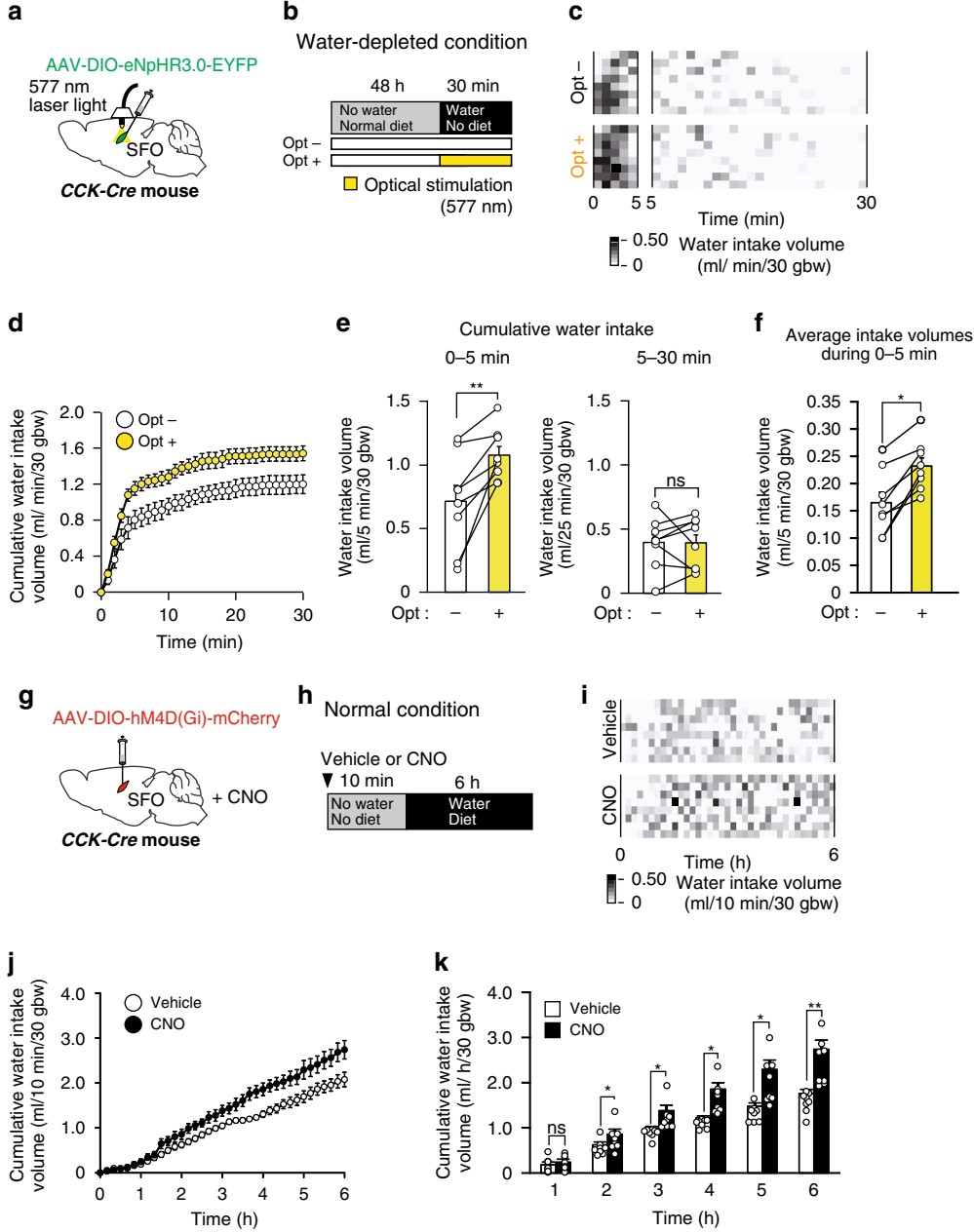

**Fig. 7 Optical and chemogenetic silencing of CCK-positive SFO neurons enhances water drinking under both water-depleted and water-repleted conditions. a** Injection of AAV-DIO-eNpHR3.0-EYFP into the SFO of the *CCK-Cre* mouse. **b** Experimental protocol to estimate water intake under the water-depleted condition with (opt+) or without (opt−) optical inactivation (wavelength, 577 nm). **c** Grayscale heat maps of water intake by individual mice with or without optical inactivation. 0–5 min (left), 5–30 min (right). **d** Water-intake volumes during 30 min with or without optical inactivation. **e** Summary of water intake under the water-depleted condition with or without the optical silencing of CCK-positive neurons in the SFO ($n = 8$ mice each; $W_{(0)} = 0$, $P = 0.008$; $W_{(5)} = 16$, $P = 0.834$). 0–5 min (left), 5–30 min (right). **f** Averages of water intake volume with or without optical inactivation during 0–5 min ($W = 0$, $P = 0.0142$). **g** Injection of AAV-DIO-hM4D(Gi)-mCherry into the SFO of the *CCK-Cre* mouse. **h** Experimental protocol to estimate water intake under the normal (free access to water and diet) condition with the subcutaneous injection of vehicle or CNO (2 mg/kg). **i** Grayscale heat maps of water intake by individual mice ($n = 8$ mice each). **j** Water-intake volumes over 6 h with the vehicle or CNO injection. **k** Summary of cumulative water intakes in mice with the vehicle or CNO injection under the normal condition ($n = 8$ mice each; $W_{(1)} = 10$, $P = 0.313$, $W_{(2)} = 36$, $P = 0.014$, $W_{(3)} = 0$, $P = 0.014$; $W_{(4)} = 0$, $P = 0.014$; $W_{(5)} = 1$, $P = 0.021$; $W_{(6)} = 0$, $P = 0.008$). Scale bar, 50 μm. bw, body weight; ns, not significant; **$P < 0.01$; *$P < 0.05$; two-sided Wilcoxon's signed-rank tests in (**e**), (**f**), and (**k**). Data represent the mean ± s.e.m.

neurons were present in the SFO and innervated nearby GABAergic interneurons expressing CCK-B receptors (Fig. 2b–e). The water intake under the dehydrated condition was reduced by the injection of CCK in the SFO (Fig. 2f–h). Our single-cell dynamics of CCK-positive neurons newly revealed that two distinct subpopulations of CCK-positive neurons exist in the SFO

(Fig. 3g): one is persistently activated under the Na-depleted condition (group 1 CCK-positive neurons), corresponding to our previous findings[25], while the other is transiently activated in response to water ingestion by the signal from the gastrointestinal tract (group 2 CCK-positive neurons). Furthermore, optogenetic and chemogenetic silencing experiments demonstrated that group

1 and group 2 CCK-positive neurons were involved in the persistent and transient suppression of thirst, respectively (Figs. 4 and 7). The activities of water neurons thus appear to be controlled by two distinct CCK-positive neurons via GABAergic interneurons in the SFO.

CCK has a long history as a gastrointestinal hormone and more recently as a neurotransmitter or neuromodulator in the brain[41–43]. Recent findings indicate that CCK modulates intrinsic neuronal excitability and synaptic transmission in a cell-type-specific manner, acting as a key molecular switch to regulate the functional output of neuronal circuits[44]. The possibility that CCK increased in the SFO under the Na-depleted condition is a peripheral origin that was ruled out because CCK in plasma was maintained at constant levels irrespective of body-fluid conditions (Fig. 1a). Moreover, a significant increase in CCK mRNA levels was detected in the SFO under the Na-depleted condition (Fig. 1f), and CCK-positive SFO neurons were activated under the same condition (Fig. 5g). Our retrograde labeling experiments indicated that a small number of CCK-positive neurons were present in the superior subnucleus of the LPBN projecting to the SFO; however, they were not activated under the Na-depleted condition (Supplementary Fig. 1). Therefore, we concluded that increased CCK levels in the SFO under the Na-depleted condition were generated in the SFO.

CCK-A and CCK-B receptors are both expressed in the SFO[45]. Ahmed et al. reported that neuropeptide CCK caused depolarization in a large population (39.0%) or hyperpolarization in a small population (13.0%) of dissociated rat SFO neurons, and that Fos proteins in the SFO were induced by CCK through CCK-B receptors[45]. In our electrophysiological study, the activation of CCK-positive neurons in the SFO led to the activation of GABAergic neurons via CCK-B receptors (Fig. 2c, d). Moreover, the suppression of water intake was caused by an injection of CCK-8s, a common ligand of CCK-A and CCK-B receptors, but not a specific agonist for CCK-A receptors, into the SFO (Fig. 2h). Our results are consistent with the finding that CCK-B receptor knockout (CCKBR-KO) mice exhibited a higher water intake than WT mice[46]. A previous study showed that the activation of CCK-B receptors in parvalbumin-positive basket cells was coupled with the activation of a transient receptor potential channel to induce depolarization in the hippocampus[47]. Taken together, these findings suggest that CCK has the potential to activate CCK-B receptor-harboring neurons.

Cancelliere et al. demonstrated using dissociated rat SFO neurons that 63% of responsive SFO neurons were depolarized by both CCK and Ang II, 25% were depolarized by Ang II only, and 12% were hyperpolarized by CCK only[48]. They demonstrated that the larger population of SFO neurons excited by circulating CCK was also activated by Ang II, and that a glucose environment affects the responsiveness of these neurons to both of these hormones, showing the ability of SFO neurons to integrate multiple metabolic and cardiovascular signals[48]. We previously revealed that water neurons and salt neurons in the SFO were both positive for AT1a and activated by Ang II to induce water intake and salt intake, respectively[25]. These two excitatory neurons may correspond to the second population of the above. In addition, GAD67-negative neurons with CCK-B receptors appear to co-exist in the SFO (Fig. 2e). This neuronal population may correspond to the third population of the above. In the present study, we showed that GABAergic neurons were activated by CCK produced by CCK-positive neurons in the SFO to suppress water neurons (Fig. 2d). This result indicates that AT1a-negative GABAergic neurons depolarized by CCK are also present in the SFO, in addition to those hyperpolarized by CCK. Thus, CCK peptides circulating or adaptively generated in situ may play distinct roles in the SFO.

Consistent with our previous findings[25], a major subset of CCK-positive SFO neurons (group 1) was activated under the Na-depleted condition (Fig. 3g). The strong activities of CCK-positive SFO neurons under the Na-depleted condition were gradually (not immediately) suppressed after salt ingestion (Fig. 5c and Supplementary Fig. 6d), suggesting that these neurons are directly or indirectly controlled by [Na$^+$] levels in body fluids. Furthermore, the optical and chemogenetic silencing experiments of these CCK-positive SFO neurons stimulated water intake despite the water-repleted condition (Fig. 4c, f), indicating that these neurons play a role in persistently suppressing water intake during the Na-depleted (water-repleted) condition until the recovery of the normal [Na$^+$] balance in body fluids.

Recent studies using fiber photometry revealed that thirst-driving neurons in the SFO and MnPO were rapidly suppressed in response to visceral sensations of water intake[31,38,49]. We herein identified the group 2 subpopulation of CCK-positive SFO neurons that transiently responded to water intake (Fig. 3g), indicating that these neurons receive the water ingestion signal from the oropharynx or gastrointestinal tract. The chemogenetic persistent silencing of group 2 CCK-positive neurons also increased naturally occurring water intake under the water- and Na-repleted condition (Fig. 7f), indicating that the activation of these neurons also suppresses water ingestion to prevent excessive water intake. As shown in Fig. 6c and Supplementary Movie 2, the majority of group 2 neurons were activated almost concomitantly with water intake, whereas some group 2 neurons had a lag time of 10–20 s between water intake and their activation. The difference in these responses may be attributed to whether these neurons are controlled by a oropharyngeal or gastrointestinal signal. Glucagon-like peptide 1 receptor (GLP1R)-positive GABAergic neurons were recently reported to inhibit thirst-driving neurons in the SFO in response to thirst-satiated signals from the gastrointestinal tract[50]. These findings strongly suggest that GLP1R-positive GABAergic neurons are controlled by group 2 CCK-positive neurons.

Previous studies indicated that the SFO receives direct projections of catecholaminergic neurons in the NTS[51,52]. Moreover, some neurons including catecholaminergic neurons in the NTS received mainly gastrointestinal signals via the vagal nerve[53,54]. These results suggest that group 2 CCK-positive neurons in the SFO receive oropharyngeal or gastrointestinal inputs caused by water ingestion through the vagal/NTS afferent pathway. It is also well known that gastric distention suppresses both feeding and drinking. It would be intriguing to speculate that group 3 CCK-positive neurons are involved in the negative feedback control of drinking based on stomach-distention signals.

It currently remains unclear whether distinct populations in water neurons receive signals from distinct CCK-positive neurons separately through different GABAergic neurons, or the two signals are integrated in the same family of water neurons in the SFO. Multiple thirst-driving signals, such as Ang II in blood and [Na$^+$] (or hypertonicity) in body fluids, are sensed at the SFO and OVLT[13,15,55], and may then be transmitted to the MnPO for the integration of these signals[25,31,49]. In the MnPO, distinct populations of glutamatergic neurons have been postulated to separately receive multiple signals from the oropharynx, gastrointestinal tract, and blood[31,38]. Our present study demonstrated that even in the SFO, an upstream nucleus of the MnPO in the signaling pathway, distinct populations of neurons already receive multiple signals separately and integrate them as a thirst suppressing signal.

The injection of CCK peptides into the cerebral ventricle reportedly reduced water intake by ~20% (ref. [56]). The injection of CCK into the SFO and optical activation of CCK-positive neurons in the SFO significantly suppressed water intake under

the water-depleted condition; however, the reduction was ~30% (~0.4 mL) of the total volume of water intake induced by water deprivation in both experiments (Fig. 2f–l). Activation of GABAergic neurons in the SFO suppressed water intake at a similar level (~0.4 mL) to the activation of CCK-positive neurons (Supplementary Fig. 2a–d). Thus, our activation of CCK-positive neurons and GABAergic neurons, only the SFO was targeted. Regarding the OVLT, a significant reduction (~70%) in water intake induced by the intracerebroventricular (icv) injection of hypertonic NaCl was observed after the ablation of the OVLT in mice[57]. We recently revealed that increases in [Na$^+$] in body fluids were sensed by two independent sensor systems in the OVLT to induce water-intake behavior: Na$_x$ channels expressed in specific glial cells, the signal from which is transmitted to nearby neurons expressing transient receptor potential vanilloid 4 (TRPV4)[57,58], and sodium/hydrogen exchanger 4 (SLC9A4) expressed in different neurons, which is coupled with acid-sensing channel 1a (ASIC1a) in the same cells[59].

Besides CCK, several antidipsogenic neuropeptides including ghrelin and atrial natriuretic peptide (ANP) have been reported[60,61]. Water intake that occurred after water deprivation was significantly inhibited by icv or intravenous injection of ghrelin and ANP[62,63]. Their target sites in the brain have been postulated to be the area postrema, NTS, or SFO[62,64]. Thus, multiple neuropeptides may be involved in the interplay between the central mechanisms for quenching thirst.

In summary, the activities of water neurons in the SFO may be controlled by blood-borne persistent signals (such as [Na$^+$], Ang II, and aldosterone) and oropharyngeal/gastrointestinal transient signals of water intake. We herein identified two groups of CCK-positive excitatory neurons in the SFO that are involved in central thirst-suppressive mechanisms (Supplementary Fig. 8). The activation of these CCK-positive neurons suppressed water intake, and in an opposite way, their inhibition induced water intake even under the water-repleted condition. Psychogenetic polydipsia causes marked overhydration, leading to hyponatremia, water intoxication, and cerebral edema[65]. As a pathogenetic factor of this symptom, abnormalities in the central angiotensin system have been proposed for excessive drinking behavior[66]. Similar symptoms may be caused by dysfunctions in the neural signaling of central antidipsogenic neuropeptides such as CCK to control thirst-driving neurons. The present results on CCK systems in the SFO provide insights into the central mechanisms underlying thirst regulation to maintain physiological body-fluid conditions.

## Methods

**Experimental animals.** All experiments with animals were performed according to protocols approved by the Institutional Animal Care and Use Committee of the National Institutes of Natural Sciences, Japan (approval numbers 17A006, 18A018, and 19A040) and the Tokyo Institute of Technology (approval number D2019013). Male wild-type C57BL/6J (CLEA Japan), *CCK-ires-Cre* (CCK-Cre)[30] (Jackson Labs stock 019021), *Gad1-GFP* (Δneo) (GAD67-GFP)[67], and *Slc32a1-ires-Cre* (Vgat-Cre)[68] (Jackson Labs stock 016962) mice were used at >8 weeks of age. Mice were housed in a temperature- and humidity-controlled room (24 ± 1 °C, 50 ± 10%) with a 12-h light-dark cycle (lights on at 8:00 a.m.), and were allowed free access to water and food (Rodent Diet CA-1, CLEA Japan) unless otherwise noted. All behavioral experiments were performed concurrently with the start of the dark cycle after animals had been housed individually for at least 3 days. None of the animals had any previous history before surgery or behavioral analyses. In the optogenetic and chemogenetic experiments, paired behavioral tests were conducted on the same group of mice.

**Reagents.** Furosemide (F4381, Sigma-Aldrich), CCK-8s (4100-V, Peptide Institute), L-365,260 (2767, Tocris), A-71623 (2411, Tocris), clozapine N-oxide (CNO) (4936, Tocris), and FluoSphere-conjugated Alexa555 (A20501MP, Thermo Fisher Scientific) were used.

**Quantification of CCK levels in plasma and the SFO.** CCK levels in plasma and the SFO were assessed as previously described[25]. Blood samples were collected by decapitation and subjected to measurements. Briefly, blood samples were collected in polypropylene tubes with 0.1% EDTA from unrestrained mice under the control, Na-depleted, or water-depleted condition. Blood plasma was obtained by centrifugation. To measure CCK levels in the SFO, SFO tissues dissected from mouse brains were homogenized in modified Ringer's solution. These tissues collected from three mice were used as one sample. CCK peptides were extracted from plasma or homogenized samples of the SFO with acetone and diethyl ether. These samples were dried in a vacuum chamber and dissolved in RIA buffer supplied with the radioimmunoassay (RIA) kit (Peninsula Laboratories, San Carlos, CA). Measurements were performed according to the instructions provided for the kit.

**Immunohistochemistry.** Mice were deeply anesthetized with tribromoethanol (Avertin) and then transcardially perfused with phosphate-buffered saline (PBS) followed by fixation with 20% formalin neutral buffer solution. Coronal sections of the brain were prepared with a vibratome (VT-1000S, Leica) or cryostat (CM 3050S, Leica) at 50 μm. To enhance the antigenicity of CCKBR, coronal sections were treated with microwave heating (98 °C, 10 min) in sodium citrate buffer (pH 6.0). After blocking with blocking buffer containing 5% normal donkey serum and 0.1% Triton X-100 in PBS at room temperature for 1 h, brain sections were incubated with primary antibodies at 4 °C for 2 days, and then washed twice in PBS. Sections were subsequently reacted with secondary antibodies at 4 °C for 1 day. The primary antibodies used in the present study were as follows: goat anti-Fos (1:500, sc-52G, Santa Cruz Biotechnology), rat anti-GFP (1:1000, 04404-84, Nacalai Tesque), rabbit anti-RFP (1:1000, 600-401-379, Rockland), mouse anti-RFP mAb Cocktail (1:500, M208-3, MBL), rabbit anti-β-galactosidase (1:1000, 55976, Cappel), goat anti-CCKBR (1:500, ab77077, Abcam), goat anti-nNOS (1:1000, ab1376, Abcam), and rabbit anti-GFAP (1:1000, Z0334, DAKO) antibodies. The secondary antibodies (Invitrogen) in the present study were used at 1:1000 dilution: donkey anti-mouse IgG Alexa Fluor 555 (A-31570), donkey anti-rabbit IgG Alexa Fluor 488 (A-21206), donkey anti-rabbit IgG Alexa Fluor 555 (A-31572), donkey anti-rat IgG Alexa Fluor 488 (A-21208), donkey anti-goat IgG Alexa Fluor 488 (A-11055), and donkey anti-goat IgG Alexa Fluor 555 (A32816). Brain sections were mounted on slides, and z-stack and tiled images were captured on a Zeiss LSM 700 confocal microscope using a ×20 objective. The ImageJ (NIH) was used for immunofluorescence quantification of these conforcal images.

**Quantification of mRNA expression of CCK in the SFO.** To measure the mRNA expression levels of CCK in the SFO, SFO tissues dissected from mouse brains were homogenized in Sepasol-RNA I Super G (Nacalai Tesque). Tissues collected from 2 mice were used as one sample. Total RNA and genomic DNA (gDNA) were extracted following the manufacturer's protocol (Nacalai Tesque), and their concentrations were measured using NanoDrop ND-1000 (Thermo Scientific). cDNA was prepared from total RNA using the PrimeScript RT reagent Kit with gDNA Eraser (Takara). qRT-PCR was performed using TB Green Premix Ex Taq$^{TM}$ II (Takara). The primers used for the amplification of *CCK* and *GAPDH* cDNAs were as follows: GAPDH: 5′-ATGGCGTTTCAAAAGGCAGTGAAG-3′ (forward) and 5′-TCTGCCATCACTGGTTGCAGGATC-3′ (reverse). CCK: 5′-GCTGATTTCC CCATCCAAA-3′ (forward) and 5′-GCTTCTGCAGGGACTACCG-3′ (reverse). Measurements were performed according to the instructions provided for the kit.

**Recombinant viral vectors.** AAV (with serotype DJ, Cell Biolabs) (>1.0 × 10$^{10}$ Genomic Copies (GC)/ml) were used for gene transfer in vivo. AAVDJ-*EF1α-DIO-mCherry* was used for the expression of mCherry, AAVDJ-*EF1α-DIO-GCaMP6f* for in vivo calcium imaging, AAVDJ-*EF1α-DIO-ChR2(H134R)-EYFP* and AAVDJ-*EF1α-DIO-eNpHR3.0-EYFP* for the optical manipulation, and AAVDJ-*hSyn-DIO-hM4D(Gi)-mCherry* for the chemogenetic manipulation. *EF1α*, elongation factor 1 alpha promoter; *hSyn*, human synapsin promoter; *DIO*, double-floxed inverted orientation.

**Brain surgery.** Mice were anesthetized by an intraperitoneal injection of a combined anesthetic consisting of medetomidine (0.75 mg/kg), midazolam (4.0 mg/kg), and butorphanol (5.0 mg/kg), and were then placed in a stereotactic frame (Narishige). After exposing the skull via a small incision, a small hole was drilled for the viral injection. Viral injections were performed as described previously with a minor modification[25]. Briefly, a pulled-glass pipette with a tip diameter of 20–40 μm was inserted into the brain, and viruses were injected with a microsyringe pump (Ultra Micro Pump III, World Precision Instruments) (AAV, 0.1 μl/min for 2 min; HiRet, 0.1 μl/min for 10 min). The coordinates for the viral injection into the SFO were as follows: relative to the bregma (anteroposterior, −0.7 mm; lateral, ±0.0 mm; ventral, +2.8 mm). After the glass micropipette was withdrawn, the skin incision was sutured, and animals were allowed to recover.

Immunohistochemical experiments were performed more than 2 weeks after the virus injection, and behavioral experiments were conducted after more than 4 weeks. Mice used for optogenetic experiments were subjected to the surgical implantation of stainless-steel cannulas (C311G, Plastics One) above the SFO more than 3 weeks after the virus injection; the cannula was fixed to the skull with a screw and dental acrylic. To infuse CCK-8s into the SFO, a guide cannula that was

handmade from a 27-G needle (Terumo) was implanted into the SFO and fixed to the skull. All stereotaxic injection sites were verified by immunohistochemistry after behavioral tests. When the virus infection to the target site was unsuccessful, behavioral data obtained from these animals were excluded from analyses.

Regarding calcium imaging, 2 weeks after the viral injection, mice were anesthetized and implanted with a ProView Lens Probe (1050-002202, Inscopix) with the assistance of a ProView implant kit (1050-002332, Inscopix). The lens was targeted to be 200–300 μm above the SFO using the following coordinates: anteroposterior, −0.7 mm; lateral, ±0.0 mm; ventral, +2.6 mm. Three weeks after lens implantation, a baseplate (1050-002192, Inscopix) was fixed above the lens using dental cement. During calcium imaging experiments, the baseplate provided an interface for attaching the miniature microscope nVoke1.0 (Inscopix), whereas a baseplate cover (1050-002193, Inscopix) was attached to the baseplate to protect the lens.

**Behavioral assays**. Fluid intake by individual mice was automatically monitored using a previously described system[25]. Mice were acclimated to the spouts providing distilled water and/or salt solution for more than 3 days before the tests. Behavioral experiments were conducted in a fasting state during the first 30 min of the dark period. Regarding the chronic inhibition test (Fig. 7g), the assessment of water intake was performed with the normal diet (Rodent Diet CA-1, CLEA Japan).

To generate "the Na-depleted condition", mice received subcutaneous injections of furosemide (a loop diuretic; 1 mg/25 g body weight) and were housed with free access to the Na-deficient diet (CLEA Diet No. 010, Clea Japan) and distilled water for 24 h. Mice were housed with free access to the normal diet without water for 48 h to generate "the water-depleted condition". Regarding calcium imaging, 3 M NaCl solution (150 μL/mice) was subcutaneously injected under the condition without water and diet to induce water drinking behavior. Concerning the injection of CCK-8s, the 27-G guide cannula was connected to a syringe pump using 33-G internal cannulas (Plastics One). The injection was performed at a speed of 0.1 μL/min for 30 min and started 5 min before the start of the behavioral test. Modified Ringer's solution containing (in mM): 140 NaCl, 2.5 KCl, 2 CaCl2, 1 MgCl2, 10 HEPES, 10 glucose, and 5 NaOH (pH 7.4 with HCl) was used as the vehicle solution. CCK-8s (1 μM) and A-71623 (1 μM) were dissolved in this solution.

In vivo photo illuminations of freely moving mice were conducted as previously described[25]. Briefly, a laser light was delivered through plastic optic fibers with an optical swivel (COME2-UFC, Lucir), which was connected to a yellow light laser (577 nm, 3000 mW, CW) (Genesis Taipan 577, Coherent) or blue light laser (445 nm, 1000 mW, CW) (KaLaser). To achieve optical silencing with eNpHR3.0, the laser output was maintained at 7–10 mW/mm$^2$ as measured at the tip of the fiber. The laser output (20 Hz) for the optical activation of ChR2 was maintained at 5–10 mW/mm$^2$ as measured at the tip of the fiber. In each test under the water- and/or Na-depleted condition, the optical stimulation was started 3 min before the start of the behavioral test.

**Calcium imaging**. Calcium fluorescence images were acquired at 15 frames per second, and 45% LED (450 nm) power using a miniature microscope from Inscopix (nVoke) and nVoke acquisition software (ver 1.3.0, Inscopix). Recording parameters were set based on pilot studies that demonstrated the least amount of photobleaching, while allowing for the sufficient detection of the calcium response. We used the CMOS camera (DFK33UX287, ARGO) for behavioral video recordings to synchronize calcium recordings. On the experimental day, the microscope was attached to the baseplate of the objective mouse, which was then allowed to acclimate in the home cage for 30 min. After the acclimation period, baseline fluorescence (Control) was recorded for 30 s. The mouse was Na-deprived using furosemide (1 mg/kg) with the Na-deficient diet or hypertonic sodium loading with a 3 M NaCl injection. Under the "Na-depleted condition" and "hypertonic sodium-loading condition", the microscope recorded fluorescence for 30 or 60 s. After acquisition, calcium recording files were spatially downsampled (factor of 2) and motion-corrected using Inscopix Data Processing software ver 1.3.0. The fluorescent traces of individual neurons were extracted from these images by PCA/ICA[69]. Regarding quantification, $\Delta F/F_0$ and z-scores were calculated from these 30-s periods of experimental conditions: $F_0$ is the mean value of the calcium signal during the control condition. The z-score calculation and K-means clustering were performed using the built-in Origin 2019 (Light Stone) function.

**Electrophysiology with SFO slices**. Electrophysiological experiments with CCK-Cre; GAD67-GFP mice were performed as reported previously[25] with minor modifications. AAV-DIO-mCherry was injected into the SFO at least 2 weeks before the recording. The brains were quickly removed and submerged for 5–10 min in ice-cold sucrose Ringer solution bubbled with 100% O2 (pH 7.3 with HCl) containing (in mM): 260 sucrose, 2.5 KCl, 10 MgSO4, 0.5 CaCl2, 5 HEPES, 10 glucose, and 5 NaOH. Slices were cut at a thickness of 350 μm and at an angle of 45° to the coronal plane with a microslicer (Pro 7, Dosaka EM), and then pre-incubated at room temperature for more than 1 h.

Slices were mounted in a recording chamber on an upright microscope (BX61WI, Olympus) and continuously perfused with modified Ringer's solution during recordings. In experiments performed under the 145 mM Na condition,

CCK was added to modified Ringer's solution. In recordings of the evoked EPSC, GFP-labeled neurons were selected using fluorescent optics, and the whole-cell configuration was obtained. In analyses of firing frequencies, cell-attached configurations were obtained in mCherry-labeled and GFP-positive neurons. All recordings were made at 33–36 °C. Inhibitors were applied to the slices more than 10 min after the start of the experiments, and the concentrations of the inhibitors were kept constant until the experiments had finished.

Patch pipettes were prepared with borosilicate glass capillaries, and filled with pipette solution containing (in mM): 140 K gluconate, 10 KCl, 2 MgCl2, 0.2 EGTA, 2 Na2ATP, 10 HEPES, and 0.1 spermine (pH 7.3 with HCl). The resistance of the electrodes was 3–7 MΩ in Ringer's solution. Data were acquired using Axopatch 200B and Axopatch 1D patch-clamp amplifiers (Axon Instruments) with the software pCLAMP, and analyzed with the software Clampfit (Axon Instruments).

**Data collection and statistical analysis**. Testing groups for behavioral cohorts were balanced by age and genotype, and the randomization of experimental groups was not performed. Intake volumes were monitored automatically, and analyses were not performed blind to the conditions of experiments. No statistical methods were used to predetermine sample sizes. Values are shown as the mean ± s.e.m. (error bars). Statistical analyses were performed by a one-way RM ANOVA, the two-sided Mann–Whitney U test, paired Student's t-tests, Wilcoxon's signed-rank tests, and a K-means clustering analysis using Origin 2019 (Light Stone). Data distribution was assumed to be normal, but this was not formally tested. Representative images were selected from more than three original candidates.

**Reporting summary**. Further information on research design is available in the Nature Research Reporting Summary linked to this article.

## Data availability
Data are available from the corresponding author upon request.

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

## Acknowledgements

We thank Dr. Y. Yanagawa (Gunma University, Japan) for GAD67-GFP mice and Dr. K. Deisseroth (Stanford University, USA) for the *hChR2(H134R)* gene and *eNpHR3.0* gene. We also thank Y. Isoshima, N. Nakanishi, and T. Hashimoto for their technical assistance, and A. Kodama and T. Tanaka for their secretarial assistance. This work was supported by MEXT/JSPS KAKENHI (Grant numbers 17H07331 and 18K14855 to T.M.; 26293043 to T.Y.H.; 24220010 and 19H05659 to M.N.), Core Research for Evolutional Science and Technology of the Japan Science and Technology Agency (CREST, JST to M. N., grant number JPMJCR1754), and the Takeda Science Foundation (to T.M.). This work was also supported by the Cooperative Study Program (20-254) of the National Institute for Physiological Sciences.

## Author contributions

T.M. designed experiments under the guidance of T.Y.H. and M.N., T.M. performed and analyzed the experiments; Kenta Kobayashi and Kazuto Kobayashi prepared the viral vectors; M.N. and T.M. wrote the manuscript.

## Competing interests

The authors declare no competing interests.
