## [Peer Review File · Nature Communications]

Reviewers' Comments:

Reviewer #1:

Remarks to the Author:

The manuscript by Matsuda et al. continues a line of investigation by these workers into cholecystinin-expressing neurons (CCK-neurons) within the subfornical organ that influence water intake in mice. They now establish convincingly that two different major populations of CCK-neurons exist within the SFO. One is activated in sodium-depleted mice and depresses water-drinking; the other population is activated during the act of drinking and also has a role in the suppression of drinking to satiation. The methods used are state-of-the-art and an impressive array of results provide bountiful evidence regarding the functions and operating mechanisms through GABAergic interneurons that express CCK type B receptors. Conclusions are supported by data. The work is novel and makes an important contribution to knowledge of brain mechanisms mediating the "switch-off" of drinking when sufficient water has been ingested. It will be of widespread interest in the field of ingestive behaviour.

However, this manuscript does contain some misleading or erroneous statements which need correction.

1. In "Introduction" , the first sentence states that "water-intake behaviour is strictly controlled by the body-fluid condition". This is incorrect, particularly for humans, where many factors such as habit, ritual, social custom play major roles in drinking-behavior. As well, the term "body-fluid condition" is nebulous and rather meaningless. The reference provided is somewhat derivative of earlier more comprehensive reviews e.g. Fitzsimons book on thirst or his Physiological Reviews and the book by Ramsay and Booth (1991) which give more comprehensive description of water-intake behaviour.
2. Intro, Line 41-42, states that over-ingestion of salt induces cardiac failure. This is misleading, because while excessive fluid and Na intake may worsen the prognosis in heart failure, it does not induce heart failure, but rather is more a consequence of it.
3. Intro, Line 48. While it is true that the recent studies (refs 12-14) demonstrate thirst driven by SFO and OVLT, this neglects numerous studies over the past 40 or more years showing the importance of these sites e.g. pioneering work of John Simpson and colleagues on SFO and Terry Thrasher, David Ramsay and others on OVLT which led the way.
4. Intro, line 52. Osmolality is not a stimulus to thirst (e.g. hyperosmolar urea or glucose do not stimulate osmoreceptors); "hypertonicity" or "cellular dehydration of the osmoreceptor" is the stimulus.
5. Why is the NeuN immunostaining so sparse and confined to the dorsal part of the SFO in Fig 1c?
6. Results. Line 91. "Water-deprivation" is more exact description of the procedure than "dehydration".
7. Results line 117. Which sub-nucleus of the LPBN contain CCK-neurons?
8. Results, line 183. The Na-deplete condition is said to be over-hydration. This is incorrect if furosemide is the agent used. Both water and Na are lost in urine, but the resulting water intake usually does not make up the urine volume loss, so in fact mice may be hyponatremic but strictly speaking they will still be in negative water balance, therefore slightly under-hydrated.
9. Results, line 194. It is difficult to know exactly what the terms "Body-fluid condition" and "Visceral water sensation" mean. In regard to the latter, it seems to imply that there is a conscious top-down influence of the sensation of drinking, but is there any evidence for this? Perhaps "water-ingestion signals from gastro-intestinal tract" would be more helpful.
10. Discussion. Do the authors have any clues as to the afferent neural inputs and signals that reach these CCK-neurons to regulate drinking? If so, it would enhance the Discussion.
11. Methods, line 832. It is stated that mice were "water-deprived with 3M NaCl Injection". This is incorrect and the mice are not "water depleted" as stated on line 833. The hypertonic saline increases plasma [Na] and tonicity, but does not deplete the animals of water, and should not be equated to the water deplete condition where both intra- and extracellular volume is reduced and Ang II (an important dipsogen) and vasopressin blood levels rise. This is an important consideration because with

hypertonic saline injection, while there is intracellular dehydration, extracellular volume does not fall and plasma Ang II levels fall to low levels as a result of positive Na balance inhibiting renin secretion from the kidney. This should also be corrected in Figure 3d and 3e (and lines 175, 177 179 in results section) where the term "Water-depleted condition" is used incorrectly when it should say "hypertonic sodium loading". The same comment applies to Fig. 6 a.

Minor corrections. Fig. 4f, right panel. Change "Vechile" to "Vehicle"

Reference list: Line 390-391 and 470, fix upper case words and Ref 25, should "International Union of Pharmacology XXI" be in the title?

Reviewer #2:

Remarks to the Author:

Matsuda et al report in this paper that separate subpopulations of CCK positive SFO neurons are potentially involved in slightly different aspects of water intake. Effectively this study represents a follow from previous reports from the authors identifying water and salt intake neurons in the SFO with different projections sites. Effectively the new aspect of this paper really is associated with the finding that CCK and CCK receptor expressing neurons may be involved in slightly different aspect of fluid intake.

The authors have used a combination of very sophisticated techniques to try to address the specific roles of these subsets of SFO neurons and provided when they drill right to precise specifics that there may be roles for these neurons in very specific components of fluid intake in the quite intense depletion models. However, I found the paper incredibly difficult to read and follow the specific questions being asked as well as the big picture identification of the potential relevance of the outcomes.

Specific questions and suggestions are outlined below:

- The introduction does not really highlight the literature associated with CCK actions in the SFO, is it expressed, are the receptors there, what are demonstrated functions in this structure if any.
- The introduction provides in my view a somewhat myopic view of the roles of the CVOs in the regulation of thirst, completely ignoring almost 40 years of history since the initial reports of the role of the SFO in the regulation of drinking and sensing of circulating angiotensin II.
- Does CCK act in SFO to directly influence fluid intake under normal or fluid/sodium depletion models, I am not aware of any data that answer this question.
- Having read the authors previous papers identifying water and sodium output neurons of the SFO. It would have been incredibly helpful to this reader to see a clear perspective of how this report integrates with their own previous work.
- The authors indicate in their report that nNOS expression is now the gold standard for the identification of excitatory neurons which I think is at best something of a stretch.
- It would have been helpful for me to understand what the authors see as the potential role of CCK in the regulation of the excitability of SFO neurons – did CCK have any effect of membrane potential of these neurons???
- The authors discuss previous data showing that subpopulations of SFO neurons are depolarized by angiotensin II and hyperpolarized by CCK, but do not mention the much larger population of these cells that were depolarized by both peptides – how would that observation fit with their data.
- The authors have utilized very sophisticated techniques to monitor calcium transients in SFO neurons in vivo and attempted to associate the observed changes in CCK positive neurons. The responses of these cells are then split into 3 different groups of neurons – presumably because they were not homogenous as hoped. While I understand the reason for doing this does it not completely undermine the whole tenet of this report ie that the CCK phenotype is profoundly important to the roles of these neurons.

Reviewer #3:

Remarks to the Author:

The authors look at how different SFO neurons regulate thirst. The current study is based on prior studies showing that GABAergic neurons innervating SFO thirst neurons are activated by CCK during Na⁺ depleted conditions – to inhibit thirst. This paper identifies the CCK-secreting excitatory neurons that act in the SFO. Ca imaging revealed 2 populations of cck⁺ neurons in the SFO: those activated under Na-depletion or transiently activated in response to water drinking. Silencing CCK⁺ neurons increased water intake under water replete conditions.

Overall, I think this manuscript includes solid and interesting findings. The findings are often occluded by ambiguities in the writing and data presentation that need to be corrected or clarified prior to publication. Although I am frustrated by the authors making claims that they do not substantiate, I feel that they need not necessarily perform experiments to substantiate all the claims they made, rather just rewrite to get rid of the extraneous claims that are not explored. After doing so I support publication of this interesting study.

Comments

1. Abstract is confusing. Where is the source of CCK vs. CCK B receptors? Not clear when reading abstract. Making the logical flow more relevant to the experiments actually performed would be better.
2. Introduction is unclear: Why is CCK introduced as a visceral hormone in feeding when the whole paper refers to central CCK. Why is there a long introduction on Na thirst neurons and water thirst neurons and how does it relate to this study? How do the CCK neurons relate to osmotic and hypovolemic thirst modalities that have previously been described – I think the information is in the introduction but is very hard to understand.
3. Fig 1c is confusing – what are the mCherry neurons in C that are not NeuN? If 90% are nNOS positive, then 1c does not make sense.
4. Why does the mCherry in 1C and 1D look so different, 1C looks membrane bound.
5. Writing line 110-112 is inaccurate, these results do not suggest that the CCK is released locally, they merely show that it can be produced locally.
6. Fig 2D/In 139 – authors state a subset of GABAergic neurons are activated by stimulating CCK neurons. What is the %? And what would be the expected % from the anatomy.
7. Supplementary fig 3a-d is a repeat of Oka (ref 22) which is fine but should be cited.
8. Fig 2F – how was the dose of CCK determined? Why no dose response? And is this dose physiologically relevant?
9. Fig 2H – reduction in water intake is comparable to that seen in activation of GABAergic neurons...but it is not complete. The Oka paper in 2015 does not quantify volume, so this is interesting. More commentary on this result is necessary, as it seems to reduce drinking but not stop it (see comment 13)
10. What % of SFO neurons are CCK neurons? And how does this compare to other characterized neuron populations in the SFO?
11. Fig 3 – There are 2 categories of neurons based on response to Na-depletion/drinking, but does that mean there are only 2 distinct CCK subpopulations as title to this section states? Fig 3 suggests there are at least 3 distinct populations. But a larger question – does one neural activity recording make these neurons a 'distinct population'. Further, what happens on subsequent days/repeat recordings of the same neurons?
12. Ln 198 – how did the authors selectively manipulate 'group 1 neurons' during this optical

silencing. It appears they inhibited all cck neurons.

13. Ln 205-206 – the increase in drinking observed is the converse of what is seen during CCK neuron activation, which is interesting. However, it only accounts for about 33% of the drinking a deprived animal would do (or conversely, in figure 2 a small reduction in drinking). Since the authors have pointed this out, it would be very interesting for them to comment on what they think is occurring/why the magnitude is only a small % of the drinking.

14. Fig 5c – Maybe I am misunderstanding what is being plotted in 5c, but the plots appear to be wrong as the dF/F at time point 4 for salt has a lot of 'purple' in fig 5b, suggesting that the mean dF/F would be below zero on some neurons.

15. Fig 6D – the means of z scores, and presenting everything as a z score makes it hard for the reader to interpret the actual levels of Ca dynamics and the underlying change in activity. I would encourage at least a supplement that does not simply show the z-score but the actual dF/F and supplemental videos showing the recording.

16. Again – how were just the group 2 neurons inhibited in this experiment? (see comment 12)

17. The authors claim that CCK acts on CCK-B receptors in the SFO in the abstract, but this claim is based on one experiment done with ex vivo physiology. Doing an in vivo experiment with a CCK-B KO or pharmacology, would raise the relevance of this to the abstract, but without such an experiment, the evidence supporting this claim is rather weak. Again, the writing occludes clarity of the actual findings the authors make.

Responses to the Reviewers' comments:

Responses to Reviewer #1:

1. In "Introduction", the first sentence states that "water-intake behaviour is strictly controlled by the body-fluid condition". This is incorrect, particularly for humans, where many factors such as habit, ritual, social custom play major roles in drinking-behavior. As well, the term "body-fluid condition" is nebulous and rather meaningless. The reference provided is somewhat derivative of earlier more comprehensive reviews e.g. Fitzsimons book on thirst or his *Physiological Reviews* and the book by Ramsay and Booth (1991) which give more comprehensive description of water intake behaviour.

We revised the Introduction according to the Reviewer's suggestion, and cited the recommended reference (p. 3, lines 39–40).

2. Intro, Line 41-42, states that over-ingestion of salt induces cardiac failure. This is misleading, because while excessive fluid and Na intake may worsen the prognosis in heart failure, it does not induce heart failure, but rather is more a consequence of it.

We replaced "... induce ..." with "... worsen the prognosis of ..." (p. 3, lines 43–44).

3. Intro, Line 48. While it is true that the recent studies (refs 12-14) demonstrate thirst driven by SFO and OVLT, this neglects numerous studies over the past 40 or more years showing the importance of these sites e.g. pioneering work of John Simpson and colleagues on SFO and Terry Thrasher, David Ramsay and others on OVLT which led the way.

We cited the studies recommended by the Reviewer (p. 3, lines 48–57).

4. Intro, line 52. Osmolality is not a stimulus to thirst (e.g. hyperosmolar urea or glucose do not stimulate osmoreceptors); "hypertonicity" or "cellular dehydration of the osmoreceptor" is the stimulus.

We changed "... osmolality ..." to "... hypertonicity ..." (p. 3, line 55).

5. Why is the NeuN immunostaining so sparse and confined to the dorsal part of the

SFO in Fig 1c?

As the Reviewer pointed out, the image of NeuN in the SFO is not adequate. Although the antibody for NeuN (MAB377B, Millipore) clearly stained other brain regions, including the cortex, it did not appear to function effectively in the SFO. We attempted to stain neurons in the SFO using an antibody cocktail of neuronal markers (MAB2300, Millipore); however, this cocktail also did not accurately detect neurons in the SFO. On the other hand, previous studies demonstrated that nNOS-positive cells largely overlapped with glutamatergic neurons in the SFO (refs. 30, 35). We also showed that the majority of CCK-positive cells were positive for nNOS in the SFO (new Fig. 1c). As another Reviewer suggested, we deleted old Fig. 1c and revised the description in the text (p. 6, lines 112–119).

6. Results. Line 91. Water-deprivation” is more exact description of the procedure than “dehydration”.

We replaced “... dehydration ...” with “...water deprivation ...” as suggested (p. 6, line 99).

7. Results line 117. Which sub-nucleus of the LPBN contain CCK-neurons?

Garfield et al. reported that CCK-positive neurons were located in the superior subnucleus of the LPBN (ref. 37). Consistent with this finding, CCK-positive neurons projecting to the SFO were also located in the superior subnucleus of the LPBN in the present study. We revised the line representing the border of the LPBN in Supplementary Fig. 1d, and added relevant descriptions (p. 7, lines 128–129; p. 42, lines 778–781).

8. Results, line 183. The Na-deplete condition is said to be over-hydration. This is incorrect if furosemide is the agent used. Both water and Na are lost in urine, but the resulting water intake usually does not make up the urine volume loss, so in fact mice may be hyponatremic but strictly speaking they will still be in negative water balance, therefore slightly under-hydrated.

Thank you for highlighting this important issue.

We deleted this description.

9. Results, line 194. It is difficult to know exactly what the terms “Body-fluid condition” and “Visceral water sensation” mean. In regard to the latter, it seems to imply that there is a conscious top-down influence of the sensation of drinking, but is there any evidence for this? Perhaps “water-ingestion signals from gastro-intestinal tract” would be more helpful.

We agree with the Reviewer and revised the manuscript accordingly (p. 11, line 216).

10. Discussion. Do the authors have any clues as to the afferent neural inputs and signals that reach these CCK-neurons to regulate drinking? If so, it would enhance the Discussion.

Previous studies indicated that the SFO receives direct projections from catecholaminergic neurons in the NTS (new refs. 55, 56). Moreover, some neurons including catecholaminergic neurons in the NTS reportedly receive gastrointestinal signals via vagal nerves (new refs. 57, 58). These results suggest that CCK-positive neurons in the SFO receive oropharyngeal or gastrointestinal signals via vagal/NTS afferent pathways.

We added this information to the revised manuscript (p. 18–19, lines 386–391).

11. Methods, line 832. It is stated that mice were “water-deprived with 3M NaCl Injection”. This is incorrect and the mice are not “water depleted” as stated on line 833. The hypertonic saline increases plasma [Na] and tonicity, but does not deplete the animals of water, and should not be equated to the water deplete condition where both intra- and extracellular volume is reduced and Ang II (an important dipsogen) and vasopressin blood levels rise. This is an important consideration because with hypertonic saline injection, while there is intracellular dehydration, extracellular volume does not fall and plasma Ang II levels fall to low levels as a result of positive Na balance inhibiting renin secretion from the kidney. This should also be corrected in Figure 3d and 3e (and lines 175, 177 179 in results section) where the term “Water-depleted condition” is used incorrectly when it should say “hypertonic sodium loading”. The same comment applies to Fig. 6 a.

We agree with the Reviewer and revised the manuscript accordingly (p. 10–11, lines 195, 199, 201, 205, 213; p33, line 671; p. 45, line 807; p. 57, lines 1016–1017, Figs. 3,

6, and Supplementary Fig. 4).

12. Minor corrections. Fig. 4f, right panel. Change “Vechile” to “Vehicle”

We corrected the spelling in Fig. 4f, right panel.

13. Reference list: Line 390-391 and 470, fix upper case words and Ref 25, should “International Union of Pharmacology XXI” be in the title?

We revised these words with upper case to lower case in new Refs. 5 and 54, and changed previous Ref. 25 to new Ref. 33.

Responses to Reviewer #2:

1. The introduction does not really highlight the literature associated with CCK actions in the SFO, is it expressed, are the receptors there, what are demonstrated functions in this structure if any.

According to the Reviewer's suggestion, we modified the Introduction (p. 4, lines 76–81) and Discussion (p. 15–16, lines 314–327).

2. The introduction provides in my view a somewhat myopic view of the roles of the CVOs in the regulation of thirst, completely ignoring almost 40 years of history since the initial reports of the role of the SFO in the regulation of drinking and sensing of circulating angiotensin II.

We altered the Introduction to discuss the roles of CVOs according to the Reviewer's suggestion (p. 3, lines 48–57).

3. Does CCK act in SFO to directly influence fluid intake under normal or fluid/sodium depletion models, I am not aware of any data that answer this question.

We did not observe any effects of the CCK-8s injection into the SFO on water intake over 30 min under the normal condition because the amount of water intake was originally small and, thus, it was difficult to detect a reduction in water intake (additional Supplementary Fig. 3c–e). However, the persistent chemogenetic suppression of CCK-positive SFO neurons significantly increased the cumulative volumes of water ingestion in 6 hrs under normal condition (Fig. 7k). Furthermore, we showed that the CCK-8s injection into the SFO decreased water intake under the water-depleted condition, in which acute and increased water ingestion was induced (Fig. 2f–h). On the other hand, CCK levels significantly increased in the SFO under the Na-depleted condition because CCK-positive SFO neurons were activated (Fig. 1b, 3g). The optical silencing of these CCK-positive neurons increased water intake, but not salt intake, under the Na-depleted condition (Fig. 2i–l and new Supplementary Fig. 5), suggesting that they are not involved in the control of salt intake.

4. Having read the authors previous papers identifying water and sodium output neurons of the SFO. It would have been incredibly helpful to this reader to see a clear perspective of how this report integrates with their own previous work.

We modified the Discussion accordingly (p. 15, lines 295–313).

5. The authors indicate in their report that nNOS expression is now the gold standard for the identification of excitatory neurons which I think is at best something of a stretch.

Oka et al. previously reported that nNOS-positive neurons were also positive for vglut2, a marker of excitatory neurons, but not GAD, a marker of inhibitory neurons, in the SFO and MnPO (new refs. 30, 35). In addition, our electrophysiological study demonstrated that GABAergic neurons were synaptically activated by CCK-positive neurons (Fig.2c). Therefore, nNOS expression has potential as a marker of excitatory neurons at least in the SFO. We added the description “...in the SFO” (p. 7, lines 116–119).

6. It would have been helpful for me to understand what the authors see as the potential role of CCK in the regulation of the excitability of SFO neurons – did CCK have any effect of membrane potential of these neurons?

A previous study indicated that CCK-B receptor signaling in parvalbumin-positive basket cells activated a transient receptor potential channel to induce depolarization in the hippocampus (new ref. 51). Ahmed et al. also reported that the neuropeptide CCK caused depolarization in a large population (39.0%) of dissociated rat SFO neurons (new ref. 49). These findings indicated that CCK has the potential to depolarize neurons through CCK-B receptors in the SFO. Consistent with these findings, GABAergic neurons in the SFO were depolarized by CCK-8s through CCK-B receptors in the present study (Fig. 2b–e). We added this information to the Discussion (p. 16, lines 328–341).

7. The authors discuss previous data showing that subpopulations of SFO neurons are depolarized by angiotensin II and hyperpolarized by CCK, but do not mention the much larger population of these cells that were depolarized by both peptides – how would that observation fit with their data.

A previous study using dissociated rat SFO neurons demonstrated that 63% of responsive SFO neurons were depolarized by both CCK and Ang II, 25% were

depolarized by Ang II only, and 12% were hyperpolarized by CCK only (new ref. 52). These findings showed that the larger population of SFO neurons excited by circulating CCK was also activated by Ang II, and that a glucose environment affects the responsiveness of these neurons to both of these hormones, indicating the ability of SFO neurons to integrate multiple metabolic and cardiovascular signals (new ref. 52). We previously revealed that water neurons and salt neurons in the SFO were both positive for AT1a, and activated by Ang II to induce water intake and salt intake, respectively (new ref. 29). These two excitatory neurons may correspond to the second population of the above. In addition, GAD67-negative neurons with CCK-B receptors appear to co-exist in the SFO (Fig. 2e). This neuronal population may correspond to the third population of the above. In the present study, we showed that GABAergic neurons were activated by CCK secreted from CCK neurons in the SFO to suppress water intake (Fig. 2d). Our results suggest the presence of AT1a-negative GABAergic neurons depolarized by CCK, in addition to those hyperpolarized by CCK in the SFO. Thus, CCK peptides circulating or adaptively generated *in situ* may play distinct roles in the SFO.

We added this information to the revised text (p. 17, lines 342–358).

8. The authors have utilized very sophisticated techniques to monitor calcium transients in SFO neurons in vivo and attempted to associate the observed changes in CCK positive neurons. The responses of these cells are then split into 3 different groups of neurons – presumably because they were not homogenous as hoped. While I understand the reason for doing this does it not completely undermine the whole tenet of this report ie that the CCK phenotype is profoundly important to the roles of these neurons.

We identified distinct types of CCK-positive neurons (groups 1~3) in the SFO by monitoring individual calcium activities. Group 1 (~58%) was persistently activated under the hyponatremic condition, while group 2 (~31%) was transiently activated in response to water intake. The two groups of CCK neurons are involved in the suppression of water intake. Group 3 (~11%) was randomly activated; however, we were unable to identify specific stimuli in the present study. Therefore, this group may have functions other than fluid intake. Further studies are needed to clarify whether Group 3 is really one population and what its functional roles are.

Responses to Reviewer #3:

1. Abstract is confusing. Where is the source of CCK vs. CCK B receptors? Not clear when reading abstract. Making the logical flow more relevant to the experiments actually performed would be better.

We revised the manuscript according to the Reviewer's suggestion (p. 2, lines 23–28).

2. Introduction is unclear: Why is CCK introduced as a visceral hormone in feeding when the whole paper refers to central CCK. Why is there a long introduction on Na thirst neurons and water thirst neurons and how does it relate to this study? How does the CCK neurons relate to osmotic and hypovolemic thirst modalities that have previously been described – I think the information is in the introduction but is very hard to understand.

We modified the Introduction according to the Reviewer's suggestion (p. 4, lines 76–81).

3. Fig 1c is confusing – what are the mCherry neurons in C that are not NeuN? If 90% are nNOS positive, then 1c does not make sense.

As suggested by the Reviewer, antibodies used for neuronal markers including anti-NeuN (MAB377B, Millipore and MAB2300, Millipore) did not function effectively in the SFO, though accurately stained other brain regions. Thus, we deleted previous Fig. 1c and revised the description (p. 6–7, lines 112–119).

4. Why does the mCherry in 1C and 1D look so different, 1C looks membrane bound.

We deleted Fig. 1c as described above.

5. Writing line 110-112 is inaccurate, these results do not suggest that the CCK is released locally, they merely show that it can be produced locally.

Our additional experiment showed a significant increase in CCK mRNA levels in the SFO under the Na-depleted condition (Fig.1f). This result indicates that CCK is produced in the SFO. Taken together with the result showing that the content of the CCK peptide is increased in the SFO, the CCK neuropeptide appears to be released in

the SFO under this condition. We added this information to the revised manuscript (p. 7, lines 120–122).

6. *Fig 2D/ln 139 – authors state a subset of GABAergic neurons are activated by stimulating CCK neurons. What is the %? And what would be the expected % from the anatomy.*

We already demonstrated that CCK-B receptor-positive GABAergic neurons accounted for $53.1 \pm 6.52\%$ of GABAergic neurons in the SFO (Fig. 2e), and we detected CCK-responsive firing activities from 6 out of the 14 GABAergic neurons tested (42.9%). We revised the manuscript accordingly (p. 8, line 151).

7. *Supplementary fig 3a-d is a repeat of Oka (ref 22) which is fine but should be cited.*

We revised the manuscript as suggested (p. 43, line 789).

8. *Fig 2F – how was the dose of CCK determined? Why no dose response? And is this dose physiologically relevant?*

In previous studies, CCK-8 was used (at $\sim 10 \mu\text{M}$ for *in vivo* experiments, into the cerebral ventricle; or 10 nM for *in vitro* experiments) (new refs. 49, 50), and the affinity threshold of CCK-B receptors was $\sim 300 \text{ pM}$ (Noble et al., *Pharmacol. Rev.* 1999.). In the present study, we selected CCK-8s concentrations (at $1.0 \mu\text{M}$ for *in vivo* experiments; $0.1 \mu\text{M}$ for *in vitro* experiments) that activated CCK-B receptors in GABAergic neurons. The injection of CCK-8s into the SFO and the optical activation of CCK-positive neurons both suppressed water intake at the same level ($\sim 0.4 \text{ mL}$) under the water-depleted condition (Fig. 2h, l). On the other hand, the optical and chemogenetic suppressions of CCK-positive neurons also increased water intake by $\sim 0.4 \text{ mL}$ under the Na-depleted condition (Fig. 4c, f). Taken all together, the dose of CCK-8s injected into the SFO in the present study appears to be appropriate.

9. *Fig 2H – reduction in water intake is comparable to that seen in activation of GABAergic neurons...but it is not complete. The Oka paper in 2015 does not quantify volume, so this is interesting. More commentary on this result is necessary, as it seems to reduce drinking but not stop it (see comment 13)*

In our experimental activations of CCK-positive neurons and GABAergic neurons, only the SFO was targeted (Fig. 2f-l and Supplementary Fig. 2a). Injection of CCK into the SFO and optical activation of CCK-positive neurons significantly suppressed water intake under the water-depleted condition; however, the reduction was approximately 30% (~0.4 mL) of the total volume of water intake induced by the water deprivation in both experiments (Fig. 2h, l). Consistent with our results, the injection of CCK peptides into the cerebral ventricle reduced water intake by ~20 % (new ref. 60).

Besides CCK, several antidipsogenic neuropeptides including ghrelin and atrial natriuretic peptide (ANP) have been reported (new refs. 61, 62). Water intake that occurred after water deprivation was significantly inhibited by icv or intravenous injection of ghrelin and ANP (new refs. 63, 64). Their target sites in the brain have been postulated to be the area postrema, NTS, or SFO (new refs. 63, 65). Thus, multiple neuropeptides may be involved in the interplay between the central mechanisms for quenching thirst. We added these descriptions to the revised manuscript (p. 19–20, lines 403–415).

10. What % of SFO neurons are CCK neurons? And how does this compare to other characterized neuron populations in the SFO?

We were unable to clarify the precise proportion of CCK-positive neurons in SFO neurons, because visualizations of CCK-positive neurons depend on the infection rate of virus vectors in each experiment. In immunohistological images, CCK-positive neurons did not overlap with AT1a or GABAergic signals (Fig. 1e, 2b), and we presumed that CCK-positive neurons were a smaller population than AT1a-positive neurons and GABAergic neurons in the SFO; however, precise ratios remain unknown.

11. Fig 3 – There are 2 categories of neurons based on response to Na-depletion/drinking, but does that mean there are only 2 distinct CCK subpopulations as title to this section states? Fig 3 suggests there are at least 3 distinct populations. But a larger question – does one neural activity recording make these neurons a ‘distinct population’. Further, what happens on subsequent days/repeat recordings of the same neurons?

We measured the neural activities of CCK-positive neurons based on the position of the neurons and performed similar experiments under different conditions on the following day (Fig. 3e and additional Supplementary Video 1, 2). CCK-positive neurons in the

SFO were grouped into at least three different subpopulations based on their responses to $[\text{Na}^+]$ in body fluids or water ingestion. In our experiments, we have an impression that similar results were obtained from different mice under the same conditions. Additionally, for example, group 2 CCK-positive neurons repeatedly responded to water ingestion during one experiment (see Supplementary Video 2).

12. Ln 198 – how did the authors selectively manipulate ‘group 1 neurons’ during this optical silencing. It appears they inhibited all cck neurons.

In the present study, optical and chemogenetic manipulations suppressed all CCK-positive neurons in the SFO under the Na-depleted condition (Fig. 4a, d), as the Reviewer pointed out. However, the Na-depleted condition specifically stimulated “group 1 neurons”, not “group 2 or 3 neurons” (Fig. 3g), and thus, the optical and chemogenetic silencing of all CCK-positive neurons under the Na-depleted condition may selectively inhibit “group 1 neurons”. We modified the manuscript accordingly (p. 11, lines 220–224).

13. Ln 205-206 – the increase in drinking observed is the converse of what is seen during CCK neuron activation, which is interesting. However, it only accounts for about 33% of the drinking a deprived animal would do (or conversely, in figure 2 a small reduction in drinking). Since the authors have pointed this out, it would be very interesting for them to comment on what they think is occurring/why the magnitude is only a small % of the drinking.

We modified the Discussion accordingly (p. 19–20, lines 403–415).

14. Fig 5c – Maybe I am misunderstanding what is being plotted in 5c, but the plots appear to be wrong as the dF/F at time point 4 for salt has a lot of ‘purple’ in fig 5b, suggesting that the mean dF/F would be below zero on some neurons.

We revised the range of Raster plots in Fig. 5b. The mean of purple signals at time point 4 for salt in Fig.5b decreased to between 0% and -1.0% in Fig.5c.

15. Fig 6D – the means of z scores, and presenting everything as a z score makes it hard for the reader to interpret the actual levels of Ca dynamics and the underlying change in activity. I would encourage at least a supplement that does not simply show the

z-score but the actual dF/F and supplemental videos showing the recording.

We added new Supplementary Fig. 5 and 7, and Supplementary Video 1 and 2, as requested.

16. Again – how were just the group 2 neurons inhibited in this experiment? (see comment 12)

As the Reviewer pointed out, all CCK-positive neurons in the SFO appeared to be suppressed by optical and chemogenetic manipulations during water intake in Fig.7a and 7g. However, water drinking specifically stimulates “group 2 neurons”, not “group 1 and 3 neurons” (Fig. 3g). Therefore, the optical and chemogenetic silencing of CCK-positive neurons during the onset of drinking indicates the specific inhibition of “group 2 neurons”. We revised the manuscript accordingly (p. 13, lines 275–277).

17. The authors claim that CCK acts on CCK-B receptors in the SFO in the abstract, but this claim is based on one experiment done with ex vivo physiology. Doing an in vivo experiment with a CCK-B KO or pharmacology, would raise the relevance of this to the abstract, but without such an experiment, the evidence supporting this claim is rather weak. Again, the writing occludes clarity of the actual findings the authors make.

The injection of A-71623, a selective agonist for CCK-A receptor, into the SFO did not influence water intake under the water-depleted condition (Fig.2f–h), again suggesting that the suppression of water intake is mainly controlled by CCK-B receptors in the SFO. We revised the manuscript accordingly (p. 9, lines 174–178).

Reviewers' Comments:

Reviewer #1:

Remarks to the Author:

The author's have revised the manuscript and have dealt with my comments and criticisms adequately. In the tracked changes copy, the reference list had an omission for the refs numbers 49 and 67. Their discovery of the different classes of CCK-containing neurons in the subfornical organ that regulate water intake is an important addition to knowledge on body fluid balance.

Signed Michael McKinley

Reviewer #2:

Remarks to the Author:

In their revised manuscript the authors have attempted to address comments in my initial review. However, in my opinion they have for the most part done this in a somewhat superficial way.

Reviewer #3:

Remarks to the Author:

I find the manuscript impressive and greatly improved. I am generally supportive of publication and am not sure why the authors push claims beyond the data. I am not sure the overwrite creates much harm, but as a reader it is unsettling and makes me question why these interpretations are being made. This is a strong set of studies and I urge the authors to parsimoniously make conclusions that are supported by the data (there are a lot and they are important).

Unfortunately, in the rebuttal letter, the line call outs that were changed do not match the text. This has made seeing the changes in response to a comment impossible and I would request these be modified before I can finish the review. I have highlighted some of my remaining concerns that came up during this read, but could not determine if some of my other concerns were properly addressed because I became confused with what the authors considered being changed.

Some remaining concerns from this version:

1. While the OVLT is clearly important for drinking/thirst, the current OVLT paragraph in the introduction does not seem to advance the narrative and is somewhat confusing given all experiments in the manuscript are on the SFO.
2. Ln 130 – (ref former review point 5) to strongly suggest the CCK is produced in the brain and relevant for the behavior, you must knock it down/out. The expression data is quite compelling, but not compelling enough to support the subsequent writing.
3. It is hard to ascribe function to a particular subpopulation in the inhibition experiment (where all neurons are inhibited). The authors claim that only type 2 neurons are active under those conditions, but that is a report from calcium dynamics, and does not mean there is no activity in the other CCK neurons. While this is treacherous, as written, I can accept the claims being made for the results. However, it is not clear why the two different types of inhibition give such different phenotypes. Both increase drinking but the chemogenetic approach does not reveal an increase for several hours.

Point-by-point responses to the Reviewers' comments:

Responses to Reviewer #1:

The author's have revised the manuscript and have dealt with my comments and criticisms adequately. In the tracked changes copy, the reference list had an omission for the refs numbers 49 and 67. Their discovery of the different classes of CCK-containing neurons in the subfornical organ that regulate water intake is an important addition to knowledge on body fluid balance.

Signed Michael McKinley

Thank you for your efforts to review our manuscript. We confirmed the reference numbers in the ref list.

Responses to Reviewer #2:

In their revised manuscript the authors have attempted to address comments in my initial review. However, in my opinion they have for the most part done this in a somewhat superficial way.

We really appreciate your review of our manuscript.

Responses to Reviewer #3:

I find the manuscript impressive and greatly improved. I am generally supportive of publication and am not sure why the authors push claims beyond the data. I am not sure the overwrite creates much harm, but as a reader it is unsettling and makes me question why these interpretations are being made. This is a strong set of studies and I urge the authors to parsimoniously make conclusions that are supported by the data (there are a lot and they are important.

Unfortunately, in the rebuttal letter, the line call outs that were changed do not match the text. This has made seeing the changes in response to a comment impossible and I would request these be modified before I can finish the review. I have highlighted some of my remaining concerns that came up during this read, but could not determine if some of my other concerns were properly addressed because I became confused with what the authors considered being changed.

In the last resubmission, we prepared the line call-outs in “a point-by-point response to referees letter file” based on the revised manuscript (modifications-omitted version). We also prepared and submitted the revised manuscript with tracked changes. Sorry, but we did not know only the latter version is sent to reviewers.

Some remaining concerns from this version:

1. While the OVLT is clearly important for drinking/thirst, the current OVLT paragraph in the introduction does not seem to advance the narrative and is somewhat confusing given all experiments in the manuscript are on the SFO.

We moved the OVLT paragraph in the introduction (p. 3, lines 58–65) to the discussion (p. 20, lines 435–443).

2. Ln 130 – (ref former review point 5) to strongly suggest the CCK is produced in the brain and relevant for the behavior, you must knock it down/out. The expression data is quite compelling, but not compelling enough to support the subsequent writing.

Our results indicated that mRNA and peptide of CCK are significantly induced in the SFO under the Na-depleted condition (Fig. 1a, f). In addition, GABAergic neurons innervated by CCK-positive excitatory neurons in the SFO are activated by CCK-8s peptide or inhibited by CCK-B receptor antagonist *in vitro* (Fig. 2c, d). Furthermore, water intake volume under the water-depleted condition were significantly reduced by injection of CCK-8s (Fig. 2f–h). These results support the view that CCK produced in the SFO is involved in the regulation of water-intake behavior as the neurotransmitter. We think these circumstantial evidences sufficiently indicate that CCK is secreted by CCK-positive neurons.

However, as the reviewer pointed out, the secretion of CCK itself has not been demonstrated by experiments. We agree with the reviewer; however, the knock down/out experiments including construction of virus vectors are time-consuming. And alternatively, we revised the descriptions regarding this point by toning down (p. 6, line 96; p. 7, line 124; p. 15, lines 314 and 326; p. 21, line 453; p. 44, line 821; p. 51–52, lines 904 and 910).

3. It is hard to ascribe function to a particular subpopulation in the inhibition experiment (where all neurons are inhibited). The authors claim that only type 2

neurons are active under those conditions, but that is a report from calcium dynamics, and does not mean there is no activity in the other CCK neurons. While this is treacherous, as written, I can accept the claims being made for the results. However, it is not clear why the two different types of inhibition give such different phenotypes. Both increase drinking but the chemogenetic approach does not reveal an increase for several hours.

We do not think that two different types of inhibition gave different phenotypes. Optogenetic method inhibits neuronal activity for a short period by activation of exogenous Cl⁻ channels only during the optical stimulation. Chemogenetic method also inhibits neuronal activity for several hours through activation of an exogenous receptor coupled to Gi, which leads to activation of G-protein activated inwardly rectifying potassium (GIRK) channel. Group 2 CCK neurons are activated by the signal of water intake. Therefore, we used different conditions for mice to take significant amounts of water in order to observe the effect of inhibition. In the former case (Fig. 7a–f), the effect was observed only for a short period with the optical stimulation. In the latter case using a drug (Fig. 7g–k), it takes a certain time to detect the effect, but the effect continues for several hours. We revised the descriptions about optogenetic and chemogenetic approaches more intelligible (p. 12, lines 239–242; p. 13–14, lines 283–304).

In figure 7a–f, we showed that optical silencing of CCK-positive neurons in the SFO increased water intake during a short time with optical stimulation (about 5 min) under water-depleted condition, in which high frequency of water intake is expected. On the other hand, in figure 7g–k, we showed that chemogenetic inhibition of CCK-positive neurons in the SFO also increased naturally occurring water intake under normal (euhydrated) condition. In this condition, water intake occurs at low frequency but we observed the effect for several hours, enough time to detect the effect. Significant increase in the cumulative water-intake volume was not detected during the 1st hour period (Fig. 7k). However, the difference in the amount of water intake became clear from the 2nd hour period (Fig. 7k).